# A review of Last Interglacial sea-level proxies in the Western Atlantic and Southwestern Caribbean, from Brazil to Honduras

Karla Rubio-Sandoval[1], Alessio Rovere[1], Ciro Cerrone[2], Paolo Stocchi[3], Thomas Lorscheid[4], Thomas Felis[1], Ann-Kathrin Petersen[1], Deirdre D. Ryan[1]

[1]MARUM - Center for Marine Environmental Sciences, University of Bremen, Germany
[2]Dipartimento di Scienze della Terra, Universitá degli Studi di Pisa, Italy
[3] NIOZ, Royal Netherlands Institute for Sea Research, Coastal Systems Department, and Utrecht University, PO Box 59 1790 AB Den Burg (Texel), The Netherlands
[4] Department of Geography - Research Group for Earth Observation (rgeo), Heidelberg University of Education, Germany

*Correspondence to*: Karla Rubio-Sandoval (krubiosandoval@marum.de)

**Abstract.** We use a standardized template for Pleistocene sea-level data to review last interglacial (MIS 5) sea-level indicators along the coasts of the Western Atlantic and Southwestern Caribbean, on a transect spanning from Brazil to Honduras, and including the islands of Aruba, Bonaire, and Curaçao. We identified six main types of sea-level indicators (beach deposits, coral reef terraces, lagoonal deposits, marine terraces, *Ophiomorpha* burrows, and tidal notches) and produced 55 standardized data points, each constrained by one or more geochronological methods. Sea-level indicators are well preserved along the Brazilian coasts, providing an almost continuous north-to-south transect. However, this continuity disappears north of the Rio Grande do Norte Brazilian state. According to the sea-level index points (discrete past position of relative sea level in space and time) the paleo sea-level values ranging from ~5.6 to 20 m a.s.l. in the continental sector, and from ~2 to 10 m a.s.l. in the Caribbean islands. In this paper, we address the uncertainties surrounding these values. From our review, we identify that the coasts of Northern Brazil, French Guyana, Suriname, Guyana, and Venezuela would benefit from a renewed study of Pleistocene sea-level indicators, as it was not possible to identify sea-level index points for the Last Interglacial coastal outcrops of these countries. Future research must also be directed at improving the chronological control at several locations, and several sites would benefit from the re-measurement of sea-level index points using more accurate elevation measurement techniques. The database compiled in this study is available in spreadsheet format at the following link: https://zenodo.org/record/5168571 (Version 1.02; Rubio-Sandoval et al., 2021).

## 1 Introduction

In this paper, we present the results of a literature survey on the Last Interglacial shorelines (here broadly defined as having formed during Marine Isotopic Stage [MIS] 5, 132–80 ka) along the Atlantic coasts of the following countries: Brazil, French Guiana, Suriname, Guyana, Venezuela, Bonaire, Curaçao, Aruba, Colombia, Panama, Costa Rica, Nicaragua, and Honduras. The area covered by this review spans a large latitudinal gradient, including a passive margin (the central-southern coasts of

Brazil) and areas located within the Caribbean Plate (Figure 1). The large geographic span of this review was selected to fill the geographic gap between the existing sea-level compilation of Simms (2021, Mexico and Northwestern Caribbean Sea) and Gowan et al. (2021, Atlantic coasts of Argentina and Uruguay).

While we found reports on Pleistocene shorelines in most countries listed above, we could only extract sea-level index points (or marine / terrestrial limiting points) for Brazil, Bonaire, Curaçao, Aruba, and for the islands of Providencia and San Andrés in Colombia (Figure 1). This was broadly caused by a lack of enough published metadata to allow a proper standardization of sea-level data for the remaining coastal areas.

We used published peer-reviewed scientific papers to compile a database of MIS 5 relative sea-level indicators using the
standardized framework of WALIS, the World Atlas of Last Interglacial Shorelines (Rovere et al., 2020). Overall, we report data contained in 36 papers, from which we extracted 50 relative sea-level (RSL) index points, 4 marine limiting, and 1 terrestrial limiting datapoint. Age constraints are associated with each geological sea-level proxy using Luminescence (n=21), U-series (n=48), and Electron Spin Resonance (ESR, n=24) dating techniques. Several outcrops were assigned minimum ages based on limiting radiocarbon ages, or other non-radiometric age constraints (e.g., chronostratigraphic
correlations). The database is available open-access at this link: https://zenodo.org/record/5168571 (Version 1.02; Rubio-Sandoval et al., 2021).

In the following sections, we first discuss the elevation measurement methods, sea-level datums, and the main dating techniques used by the original authors. We then describe the sea-level indicators identified in this work by region. We then discuss data quality, processes causing departures from eustasy, and the presence in our area of interest of sea-level index
points associated with older Pleistocene interglacials or the Holocene. Finally, we discuss potential future research directions that may be required to improve the quantity and quality of the data contained in our review.

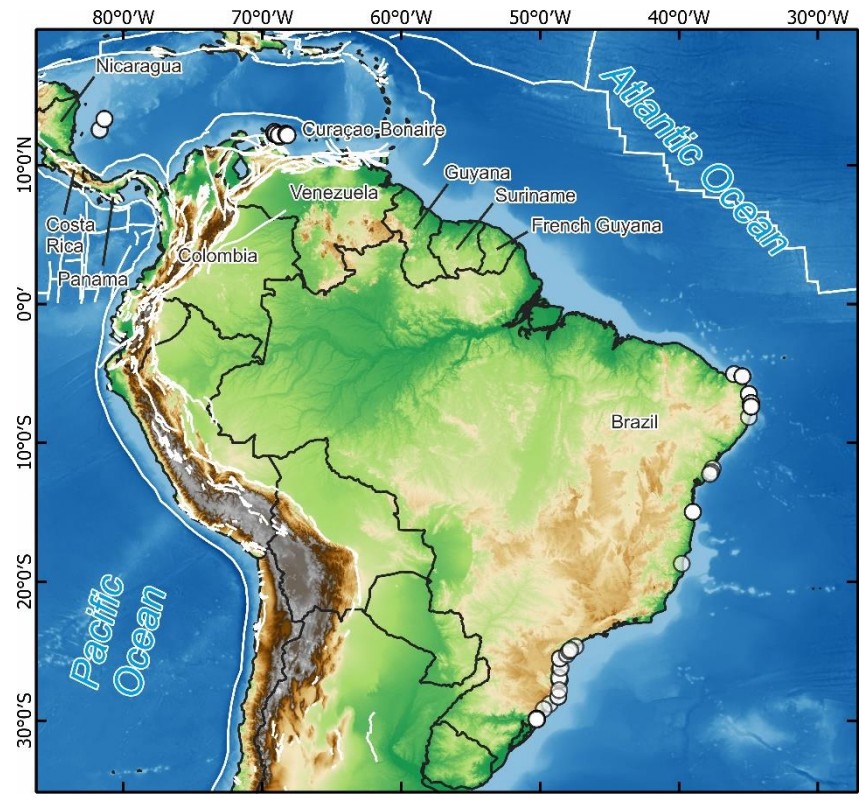

**Figure 1. General overview of the areas surveyed in this study. Dots show the location of sea-level datapoints inserted in WALIS.**
**White lines indicate the location of active faults and tectonic plate boundaries from the Global Active Faults Database (Styron and**
**Pagani, 2020) Basemap compiled by Terrestris (www.terrestris.de), with data from GEBCO (doi:10.5285/836f016a-33be-6ddc-e053-**
**6c86abc0788e), SRTM 30 m by NASA EOSDIS Land Processes Distributed Active Archive Center (LP**
**DAAC, https://lpdaac.usgs.gov/) and Natural Earth (http://www.naturalearthdata.com).**

## 2 Types of sea-level indicators

Within the region of interest (Figure 1), we identified six main types of sea-level indicators (Table 1). In addition to these,
cheniers of possible last interglacial (LIG) age are reported in French Guiana and Suriname, and beach ridges were described
in Venezuela. However, these latter instances were not included in the database due to an overall lack of sufficient information
to produce standardized index points at these locations. To correlate each point with a paleo relative sea level, we applied the
concept of indicative meaning that was introduced for Holocene sea-level studies (Shennan 1982, 1986, 1989, Shennan et al.,
1983) and recently adopted also for Pleistocene sea-level index points (Rovere et al., 2016). The indicative meaning "*describes*
*the central tendency (reference water level) and 2-sigma vertical range (indicative range) of the indicator's distribution*
*relative to tidal levels*" (Khan et al., 2019). In the database, we calculated the reference water level (RWL) and indicative range
(IR) for each sea-level index point using either modern analogs (when reported by the original study) or applying *ex-situ*
quantifications derived from global wave and tide atlases, through the IMCalc tool (Lorscheid and Rovere, 2019).

In Brazil, marine terraces are the most widespread indicator type, although fossil beaches are also common. At several locations in Brazil, sandy sediments are characterized by the presence of the *Ophiomorpha* burrows ichnofacies (Barbosa et al., 1986; Bittencourt et al., 1979; Tomazelli and Dillenburg, 2007). The main observed ichnospecies is *Ophimorpha nodosa*, which through comparison with burrows left by modern counterparts (*Callianassa major*, Frey et al., 1978), is considered an excellent sea-level indicator. Tomazelli and Dillenburg (2007) indicate that *Ophiomorpha* burrows allow the definition of the average low tide level during the deposition. However, we decided to adopt a more conservative indicative meaning (MSL to -4 m) as Frey et al. (1978) reports that, depending on the geographic region and environmental conditions, the burrows can also be found in shallow subtidal environments.

Several authors described coral reef terraces for the islands of Bonaire, Curaçao, and Aruba, located offshore Venezuela (Alexander, 1961; Schubert and Szabo, 1978; Schellmann et al., 2004). Several sea-level indicators, derived from coral reef terraces, are well preserved on Bonaire and Curaçao islands (Muhs et al., 2012; Felis et al., 2015; Obert et al., 2016; Lorscheid et al., 2017). Coral reef terraces are also reported in the Colombian islands of Providencia and San Andrés (Geister, 1972; Geister, 1986), located in the Caribbean Sea, offshore Nicaragua. At both islands, the whole reef complex is subdivided into different units according to their topography and ecology (Geister, 1992).

**Table 1. Sea-level indicators in the study area, with Reference Water Level (RWL) and Indicative range (IR) quantifications. db=breaking depth; ld=lagoonal depth; MHHW=Mean Higher High Water; MLLW=Mean Lower Low Water; MSL=Mean Sea Level; Ob=Ordinary berm; SWSH=Storm Wave Swash Height.**

| Name of RSL indicator | Description of RSL indicator | RWL | Description of IR | Indicator reference(s) |
|---|---|---|---|---|
| Beach deposit or beachrock | Definition by Mauz et al., 2015: "*Beachrocks are lithified coastal deposits that are organized in sequences of slabs with seaward inclination generally between 5° and 15°*". | (Ob + db)/2 | Ob to db | Mauz et al., 2015 Rovere et al., 2016 |
| Coral reef terrace (general definition) | Coral-built flat surface, corresponding to the area between shallow-water reef terrace and reef crest. The definition of indicative meaning is derived from Rovere et al., 2016, and it represents the broadest possible indicative range, that can be refined with information on coral living ranges. | (MLLW + db)/2 | MLLW to db | Rovere et al., 2016; Lorscheid and Rovere, 2019 |

| | | | | |
|---|---|---|---|---|
| Lagoonal deposit | Lagoonal deposits consist of silty and/or clayey sediments, horizontally laminated (Zecchin et al., 2004) and associated with fossils of brackish or marine water fauna. | (MLLW + ld)/2 | MLLW to ld | Rovere et al., 2016 Zecchin et al., 2004 |
| Marine Terrace | Definition by Pirazzoli et al., 2005: "*Any relatively flat surface of marine origin*". | (SWSH + db)/2 | SWSH to db | Pirazzoli, 2005 Rovere et al., 2016 |
| *Ophiomorpha* burrow | *Ophiomorpha* is an ichnogenus that includes burrow structures built on sandy substrates extending from MSL down to 2 m to 4 m below the surface where they divide into horizontal and inclined galleries. The burrows present a broad spectrum of morphologies and environmental distributions, mostly developing in intertidal to very shallow waters. | MSL | MSL to -4m | Frey et al., 1978 Martins et al., 2018 |
| Tidal notch | Definition by Antonioli et al., 2015: "*Indentations or undercuttings cut into rocky coasts by processes acting in the tidal zone (such as tidal wetting and drying cycles, bioerosion, or mechanical action)*". | (MHHW + MLLW)/2 | MHHW to MLLW | Antonioli et al., 2015 Rovere et al., 2016 |

## 3 Positioning and sea-level datums

In general, the majority of studies we reviewed do not report how elevations were measured (Figure 2a). Whenever this was the case, we assumed an elevation measurement error equal to 20% of the elevation reported (Rovere et al., 2016). This was also done when elevations were derived from cross-section drawings in the original publications. Other elevation measurement methods include differential GPS, metered tape or rod, topographic map, and total station (Figure 2a). Among these, the most accurate technique is differential GPS, used by Tomazelli and Dillenburg (2007) and Martins et al. (2018) in Brazil to report the elevations for the maximum height of *Ophiomorpa* burrows. Differential GPSs were also used by Muhs et al. (2012) and Lorscheid et al. (2017) in Curaçao and Bonaire, respectively. In this study, we also report new differential GPS elevation measurements taken by A. Rovere, T. Felis, and T. Lorscheid in 2016 on Bonaire at the same sites reported in Felis et al. (2015) and Obert et al. (2016). In northern Brazil, Suguio et al. (2011) used a total station to measure the elevation of different outcrops and referred the measurements to mean sea level using the tide-table predictions from two local stations. This technique also offers a good degree of absolute elevation accuracy (±0.1m/±0.2 m), depending on the reference point and its distance from the base station. The rest of the elevation measurement methods reported have different degrees of precision depending on the reference object scale.

In Brazil, elevations were referred to different datums, such as mean low water springs (Tomazelli and Dillenburg, 2007), the local geoid (Martins et al., 2018), and local sea-level datums such as the "*Brazilian Córrego Alegre National datum*" (Suguio et al., 2011). Similarly, in Curaçao, a variety of datums are used: high tide level (Schellmann et al., 2004), CARIB97, and the locally resolved geoid for the Caribbean Sea (Muhs et al, 2017). The new GPS measurements from Bonaire reported for the first time in this paper, are referred to the EGM 2008 geoid (Figure 2b;Table 2).

To obtain the geographic coordinates of the sites, in several cases, it was necessary to use Google Earth or to geocode location names to gather latitude and longitude values. Relatively few studies provided site coordinates (Figure 2c). Hence, we remark that, for some sites, the coordinates are to be interpreted as merely indicative of the general location of the site.

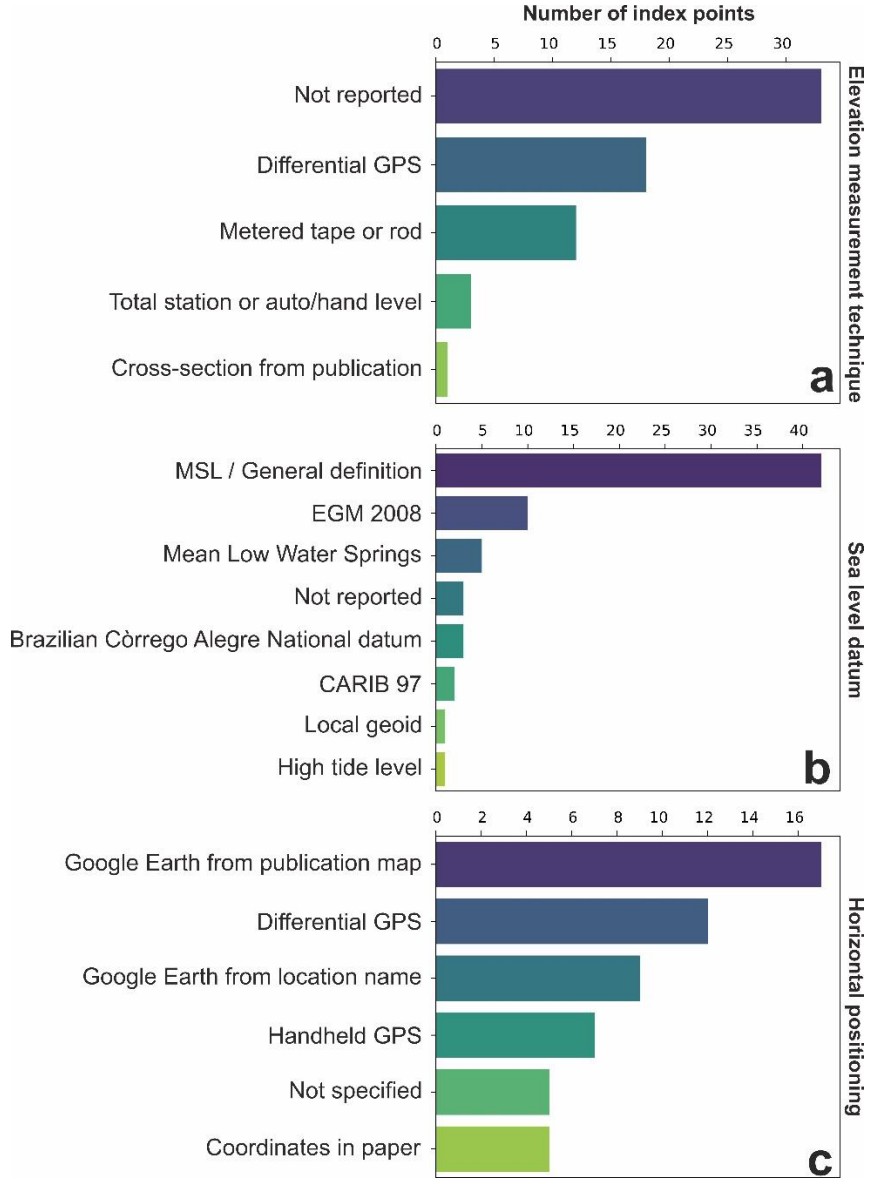


**Figure 2. Frequency of Elevation measurement techniques (a), Datums (b), and Horizontal positioning techniques (c) used in the database.**

**Table 2. Elevation datums**

| Datum name | Datum description | Datum uncertainty | Reference(s) |
|---|---|---|---|
| Brazilian Córrego Alegre National datum | Local datum for Brazil | Not available | Suguio et al., 2011 |
| CARIB97 | From Smith and Small, 1999: "*A 2×2 arc-minute resolution geoid model, CARIB97, has been computed covering the Caribbean Sea. The geoid undulations refer to the GRS-80 ellipsoid, centered at the ITRF94 (1996.0) origin*". | From the original source: "*Comparison of CARIB97 geoid heights to 31 GPS/tidal (ITRF94/local) benchmarks shows an average offset (h–H–N) of 51 cm, with a Root Mean Square (RMS) of 62 cm about the average*". | Smith and Small, 1999 |
| EGM 2008 | From Pavlis et al., 2012: "*EGM2008 is a spherical harmonic model of the Earth's gravitational potential*". | From Pavlis et al., 2012: "*Over areas covered with high quality gravity data, the discrepancies between EGM2008 geoid undulations and independent GPS/Leveling values are on the order of ±5 cm to ±10 cm*". | Pavlis et al., 2012 |
| High Tide Level | Described by Kennedy et al (2007) as the swash limit and the extent of fixed biological indicators, such as molluscs, having a restricted vertical range. | Per Rees-Jones et al (2000), accurate to +/- 2 m up to 15 m a.h.s.l and +/-5-10 m above 15 m a.h.s.l. Uncertainty will be dependent upon measurement method. | Kennedy et al., 2007 Rees-Jones et al 2000 |
| Local geoid | Geoid calculated ad hoc for the surveyed area. | Usually, very accurate. Few centimeters. | |
| Mean Low Water Springs | From Baker and Watkins (1991): "*The average of the heights ... of each pair of successive low waters during that period of about 24 hours in each semi-lunation (approximately every 14 days), when the range of the tide is greatest*". | Declared +/- 0.1 m if datum is derived from 1 year and +/- 0.25 m if measured over 1 month. | Baker and Watkins, 1991 |
| Mean Sea Level / General definition | General definition of MSL, with no indications on the datum to which it is referred to. | A datum uncertainty may be established on a case-by-case basis. | |
| Not reported | The sea-level datum is not reported and impossible to derive from metadata. | N/A | |

## 4 Dating techniques

The last interglacial deposits in the Western Atlantic and Southwestern Caribbean have been dated using a wide variety of techniques: U-series, optically stimulated luminescence (OSL), thermoluminescence (TL), electron spin resonance (ESR), chronostratigraphy, and radiocarbon dates providing minimum ages. The abundance and preservation of corals on the islands

Curaçao and Bonaire allow the application of U-series to provide reliable age assignments (Muhs et al., 2012; Felis et al., 2015; Obert et al., 2016; Lorscheid et al., 2017). ESR has also been used by Schellmann et al. (2004a) to date the coral reef terraces of Curaçao mainly to confirm the age of LIG deposits previously dated with U-series by Schubert and Szabo (1978) and to present for the first-time radiometric ages for older deposits.

Fossil corals have been preserved only at one site along the Brazilian coast. OSL and TL techniques have been used in different

outcrops in northern and southern Brazil to assess the chronology of Pleistocene marine terraces (Poupeau et al., 1988; Barreto et al., 2002; Suguio et al., 2003;2011; Buchmann and Tomazelli, 2003; Giannini et al., 2007).

Other age assignments rely on relative dating based on the cross-correlation of outcrops due to the stratigraphic similarities or radiocarbon dates (e.g., Bittencourt et al., 1979; Suguio et al., 1982; Angulo et al., 2002). When an outcrop had only minimum radiocarbon ages (radiocarbon ages above detection limit), the deposit is considered older than Holocene. There are three

chronostratigraphic constraints in Brazil (Barrier III in the Rio Grande do Sul state; Cananéia Formation in São Paulo state and Sector I deposits in Bahia state, see regional descriptions in Section 5 for details) and one in Bonaire (Lower Terrace). In Brazil, the chronostratigraphic constraints are common due to the preservation of Pleistocene deposits and their almost uninterrupted extension along the coastal plain. The Barrier III, Cananéia Formation, and Sector I deposits have been dated using OSL, TL, or U-series techniques and were attributed broader MIS 5 or more detailed MIS 5e ages (Tomazelli and

Dillenburg, 2007; Suguio et al., 2003; Martin et al.,1982).

## 5 Relative sea-level data

In the following sections, we describe the sea-level indicators in our database divided by country and, where applicable, by lower administrative boundaries (e.g. states, regions, provinces). An overview of the sites, the correlated paleo RSL, and the chronological attribution associated with them are reported in Table 3. We refer to the sea-level indicators included in the

database with their WALIS RSL ID number, shortened here as RSL ID. This number is included in the first column of the "RSL proxies" spreadsheet within the database and is a unique ID attributed automatically to each data point entered into WALIS.

Table 3. Proxies compiled in this study. Elevation and paleo RSL errors are reported as 1-sigma, age uncertainties (where an absolute age is indicated) are 2-sigma. Type of datapoints: SLI = Sea-level indicator; TLI = Terrestrial limiting, MLI= Marine limiting. Dating techniques are abbreviated as follows: LUM = Luminescence; STRAT = Chronostratigraphic constraints; Other = Other age attribution, U/Th = U-series; ESR = Electron Spin Resonance. * = ages recalculated by Chutcharavan and Dutton (2020). For the U-series ages on Bonaire, Brocas et al., 2016 reports the following average ages: [1] BON-5-A = 121±1.1 ka; [2] BON-26-A = 124.9±1.9 ka; [3] BON-24-AII.2 = 125.5±2.4 ka; [4] BON-12-A = 123.9±1.3 ka.

| WALIS RSL ID | Latitude Longitude | Site | Nation (Region) | Type of datapoint | Elevation (m) | Paleo RSL (m) | Dating technique (published sample ID) | Age (ka) |
|---|---|---|---|---|---|---|---|---|
| 154 | -29.928 -50.222 | Osorio Outcrop 04 | Brazil (R.G. do Sul) | SLI | 5.37±0.6 | 5.62±0.65 | LUM (RMG-04B) | >85.1 ka |
| | | | | | | | STRAT | MIS 5 |
| 155 | -29.923 -50.259 | Osorio Outcrop 05 | Brazil (R.G. do Sul) | SLI | 7.72±0.5 | 7.97±0.55 | LUM (RMG-04B) | >85.1 ka |
| | | | | | | | STRAT | MIS 5 |
| 153 | -29.903 -50.233 | Osorio Outcrop 03 | Brazil (R.G. do Sul) | SLI | 7.27±1 | 7.52±1.03 | LUM (RMG-04B) | >85.1 ka |
| | | | | | | | STRAT | MIS 5 |
| 152 | -29.88 -50.233 | Osorio Outcrop 02 | Brazil (R.G. do Sul) | SLI | 5.73±0.7 | 5.98±0.74 | LUM (RMG-04B) | >85.1 ka |
| | | | | | | | STRAT | MIS 5 |
| 143 | -29.862 -50.247 | Osorio Outcrop 01 | Brazil (R.G. do Sul) | SLI | 5.13±0.7 | 5.38±0.74 | LUM (RMG-04B) | >85.1 ka |
| | | | | | | | STRAT | MIS 5 |
| 1288 | -29.193802 -49.754044 | Vila Conceição | Brazil (Santa Catarina) | SLI | 9±5 | 9.18±5.1 | STRAT | MIS 5 |
| 1286 | -28.813156 -49.311895 | Coqueiros | Brazil (Santa Catarina) | SLI | 12±2 | 13±2.23 | STRAT | MIS 5 |
| 1300 | -28.285437 -48.698703 | Guaiúba | Brazil (Santa Catarina) | TLI | 6.5±1 | - | LUM (G(E)6) | 129.1±15 |
| | | | | | | | LUM (G(E)7) | 103.5±11.7 |
| 178 | -27.8188 -48.6341 | Pinheira | Brazil (Santa Catarina) | SLI | 4.5±0.2 | 6.5±2 | STRAT | MIS 5 |
| 1296 | -27.096637 -48.617834 | Itapema | Brazil (Santa Catarina) | SLI | 7.35±1 | 7.32±1.81 | STRAT | MIS 5 |
| 1297 | -27.074291 -48.597176 | Itapema Plaza Hotel | Brazil (Santa Catarina) | SLI | 7.55±1 | 7.52±1.81 | STRAT | MIS 5 |
| 1298 | -26.922263 -48.643516 | Itajaí South of Cemetery | Brazil (Santa Catarina) | SLI | 8.68±1 | 8.64±1.8 | STRAT | MIS 5 |
| 1287 | -26.327691 -48.594331 | Tapera | Brazil (Santa Catarina) | SLI | 11.5±4.6 | 11.62±4.75 | STRAT | MIS 5 |
| 179 | -26.2133 -48.52361 | São Francisco do sul Island | Brazil (Santa Catarina) | SLI | 13.5±3.53 | 13.58±3.69 | STRAT | MIS 5 |
| 181 | -25.537397 -48.578236 | Areal das Ilhas III P 01.06.05 | Brazil (Parana) | SLI | 5.5±1 | 7.5±2.23 | Other (CENA-1070) | >MIS 1 |
| | | | | | | | Other (CENA-121) | >MIS 1 |
| 186 | -25.245 -48.0783 | Canal do Varadouro | Brazil (Parana) | MLI | 4.8±1 | - | Other (CENA-121) | > MIS 1 |
| 202 | -25.00388 -47.919722 | Cananéia Island | Brazil (Sao Paulo) | SLI | 7.5±2.06 | 7.25±2.33 | STRAT | MIS 5 |
| 203 | -24.92055 -47.8275 | Comprida Island | Brazil (Sao Paulo) | SLI | 5±1 | 4.75±1.48 | STRAT | MIS 5 |
| 1299 | -24.679948 -47.464543 | Icapara | Brazil (Sao Paulo) | SLI | 9.28±1 | 9.06±1.56 | STRAT | MIS 5 |
| 204 | -18.711944 -39.804166 | São Mateus | Brazil (Esp. Santo) | SLI | 8.5±1 | 8.11±1.7 | STRAT | MIS 5e |
| 168 | -14.98 | Fazenda Jariri | | SLI | 1.27±1 | 7.77±3.64 | U/Th (CP-2) | 116±6.9 |

| | | | | | | | U/Th (CP-1) | 122±6.1 |
|---|---|---|---|---|---|---|---|---|
| | -39.003333 | | Brazil (Bahia) | | | | U/Th (CP-8) | 124±8.7 |
| | | | | | | | U/Th (CP-6) | 132±9 |
| | | | | | | | U/Th (CP-7) | 142±9.7 |
| 171 | -12.25 -37.779 | Subaúma | Brazil (Bahia) | SLI | 7.3±1 | 7.27±2.36 | STRAT | MIS 5e |
| 170 | -12.115 -37.685 | Palame | Brazil (Bahia) | SLI | 7.3±1 | 7.27±2.36 | STRAT | MIS 5e |
| 169 | -11.851 -37.577 | Conde | Brazil (Bahia) | SLI | 7.3±1 | 7.27±2.36 | STRAT | MIS 5e |
| 218 | -8.14555 -34.9708 | Lagoa Olhos-d'Agua Boa Viagem | Brazil (Pernambuco) | SLI | 10.39±2.23 | 10.38±3.15 | STRAT | MIS 5e |
| 222 | -7.396035 -34.805984 | Pitimbu beach PB17 | Brazil (Paraiba) | SLI | 5.6±1.5 | 5.53±2.41 | LUM (PB17A) | 101±9 |
| | | | | | | | LUM (PB17A) | 100±11 |
| | | | | | | | LUM (PB17B) | 71±7.7 |
| | | | | | | | LUM (PB17B) | 46±4 |
| 221 | -7.140833 -34.80861 | Cabo Branco PB10 | Brazil (Paraiba) | SLI | 9.8±1.5 | 11.8±2.5 | LUM (PB10A) | 108±8 |
| | | | | | | | LUM (PB10A) | 110±20 |
| | | | | | | | LUM (PB10B) | 138±5 |
| | | | | | | | LUM (PB10B) | 120±2 |
| 220 | -6.490277 -34.969722 | Cordosas beach PB7 | Brazil (Paraiba) | SLI | 9±1.5 | 8.99±2.64 | LUM (PB07B) | 88.9±6 |
| | | | | | | | LUM (PB07B) | 70.3±5 |
| | | | | | | | LUM (PB07C) | 110±6.2 |
| | | | | | | | LUM (PB07C) | 86±5 |
| 163 | -5.213 -35.433 | Touros outcrop | Brazil (R.G. do Norte) | SLI | 20±2 | 19.99±3.03 | LUM (32-98) | 117±10 |
| | | | | | | | LUM (32-98) | 117±10 |
| | | | | | | | LUM (39-98) | 110±10 |
| 144 | -5.056 -36.043 | São Bento outcrop | Brazil (R.G. do Norte) | SLI | 20±2 | 19.88±2.99 | LUM (32-98) | 117±10 |
| | | | | | | | LUM (32-98) | 117±10 |
| 532 | 12.052154 -68.747586 | Oostpunt | Curaçao | MLI | 3.25±0.99 | - | ESR (K4010) | 112±9 |
| | | | | | | | ESR (K4011) | 111±10 |
| 537 | 12.155712 -68.82698 | Boca Grandi | Curaçao | SLI | 5.5±1.2 | 6.45±1.46 | ESR (K4040) | 120±13 |
| | | | | | | | ESR (K4042) | 117±9 |
| | | | | | | | ESR (K4043) | 116±12 |
| 3553 | 12.235586 -69.104427 | Punta Halvedag | Curaçao | SLI | 10±0.95 | 10.98±1.28 | U/Th (Cur-Dat-16) | 124.8±0.7* |
| | | | | | | | U/Th (Cur-Dat-17) | 129.7±0.6* |
| | | | | | | | U/Th (Cur-Dat-17-A dup) | 131.4±1* |
| | | | | | | | U/Th (Cur-Dat-17-A) | 133.8±0.6* |
| 536 | 12.157296 -68.829802 | Boca Labadera | Curaçao | SLI | 5.5±1.2 | 6.45±1.46 | ESR (K4036) | 112±9 |
| | | | | | | | ESR (K4037) | 120±13 |
| 535 | 12.262312 -69.042612 | Boca San Pedro | Curaçao | SLI | 6.5±1.39 | 7.55±1.67 | ESR (K4029) | 117±12 |
| | | | | | | | ESR (K4031) | 118±80 |
| | | | | | | | ESR (K4032) | 103±11 |
| | | | | | | | ESR (K4030) | 124±13 |
| 3563 | 12.277474 -69.051679 | Boca Ascension | Curaçao | MLI | 10±2 | - | ESR (K4003) | 124±8 |
| 3554 | 12.339078 -69.153554 | Knipbai | Curaçao | MLI | 3±1.37 | - | U/Th (Cur-Dat-5) | 124±0.5* |
| | | | | | | | U/Th (Cur-Dat-5-P) | 127±0.6* |
| 533 | 12.373829 -69.125355 | Boca Cortalein | Curaçao | SLI | 10±2 | 11.05±2.2 | U/Th (Cur-Dat-1-A) | 128±1.1* |
| | | | | | | | U/Th (Cur-Dat-1) | 128.1±0.9* |
| | | | | | | | ESR (K4019) | 123±9 |

| | | | | | | | | |
|---|---|---|---|---|---|---|---|---|
| | | | | | | | ESR (K4020) | 125±11 |
| | | | | | | | ESR (K4021) | 118±8 |
| 3559 | 12.378402 -69.132382 | Boca Mansalina | Curaçao | SLI | 7.5±1.58 | 8.55±1.83 | U/Th (Cur-Dat-4) | 126.4±0.6* |
| | | | | | | | U/Th (Cur-Dat-4 dup) | 126.7±0.8* |
| | | | | | | | ESR (K4049) | 118±9 |
| 534 | 12.385825 -69.141518 | Dos Bocas | Curaçao | SLI | 10±2 | 11.05±2.2 | ESR (K4024) | 120±11 |
| | | | | | | | ESR (K4025) | 116±19 |
| | | | | | | | ESR (K4026) | 113±10 |
| 531 | 12.387517 -69.144038 | Un Boca | Curaçao | SLI | 10±2 | 11.05±2.2 | U/Th (Cur-33-d) | 118.8±0.8* |
| | | | | | | | U/Th (Cur-32) | 128±7 |
| | | | | | | | U/Th (Cur-33) | 127±7 |
| | | | | | | | U/Th (Cur-32-d) | 133.1±0.8* |
| | | | | | | | ESR (K4006) | 116±11 |
| | | | | | | | ESR (K4007) | 124±11 |
| | | | | | | | ESR (K4009a) | 140±9 |
| | | | | | | | ESR (K4009b) | 108±8 |
| | | | | | | | ESR (K4009b1) | 101±7 |
| 694 | 12.156163 -68.207258 | South of Boca Washikemba | Bonaire | SLI | 5.19±0.28 | 6.22±0.95 | U/Th (BON-5-A, Bulk)[1] | 120±1.8 |
| | | | | | | | U/Th (BON-5-A, Theca)[1] | 118.9±2 |
| | | | | | | | U/Th (BON-5-A, Bulk)[1] | 122.5±1.7 |
| | | | | | | | U/Th (BON-5-A, Theca)[1] | 121.3±1.8 |
| | | | | | | | U/Th (BON-5-A, Theca)[1] | 120.1±2.4 |
| | | | | | | | U/Th (BON-5-A, Theca)[1] | 119.4±2.7 |
| | | | | | | | U/Th (BON-5-D) | 117.7±0.8 |
| 1369 | 12.20234 -68.310734 | Notch 1 | Bonaire | SLI | 6.66±0.18 | 6.66±0.31 | STRAT | MIS 5e |
| 1370 | 12.204183 -68.312796 | Notch 2 | Bonaire | SLI | 6.61±0.11 | 6.61±0.28 | STRAT | MIS 5e |
| 1371 | 12.206776 -68.316292 | Notch 3 | Bonaire | SLI | 6.96±0.15 | 6.96±0.3 | STRAT | MIS 5e |
| 1372 | 12.2104 -68.321163 | Notch 4 | Bonaire | SLI | 6.83±0.15 | 6.83±0.3 | STRAT | MIS 5e |
| 1373 | 12.211271 -68.323699 | Notch 5 | Bonaire | SLI | 7.21±0.17 | 7.21±0.31 | STRAT | MIS 5e |
| 1374 | 12.215117 -68.335901 | Notch 6 | Bonaire | SLI | 7.26±0.12 | 7.26±0.28 | STRAT | MIS 5e |
| 693 | 12.237155 -68.285762 | Boca Olivia | Bonaire | SLI | 8.84±0.27 | 9.87±0.95 | U/Th (BON-26-A, Theca)[2] | 126.1±2.3 |
| | | | | | | | U/Th (BON-24-AII.2 Bulk)[3] | 126.7±0.97 |
| | | | | | | | U/Th (BON-24-AII.2 Theca)[3] | 122.6±1.9 |
| | | | | | | | U/Th (BON-26-A, Theca)[2] | 124.2±1.5 |
| | | | | | | | U/Th (BON-24-AII.2 Theca)[3] | 125.9±1.8 |
| | | | | | | | U/Th (BON-24-AII.2 Bulk)[3] | 128.2±2 |

| | | | | | | | |
|---|---|---|---|---|---|---|---|
| 692 | 12.247984 -68.296485 | South of Boca Onima | Bonaire | SLI | 5.72±0.3 | 6.75±0.96 | U/Th (BON-17-AI, Theca) | 121.72±0.91 |
| | | | | | | | U/Th (BON-17-AI, Theca) | 122.4±1.7 |
| | | | | | | | U/Th (BON-17-AI, Theca) | 124.2±1.8 |
| | | | | | | | U/Th (BON-17-AI, Theca) | 124.9±2.2 |
| | | | | | | | U/Th (BON-12-A, Bulk)[4] | 124.68±0.98 |
| | | | | | | | U/Th (BON-12-A, Theca)[4] | 122±1.6 |
| | | | | | | | U/Th (BON-13-AI.1, Theca) | 125.8±1.6 |
| | | | | | | | U/Th (BON-12-A, Bulk)[4] | 124.8±1.6 |
| | | | | | | | U/Th (BON-12-A, Theca)[4] | 123.6±1.6 |
| 3472 | 12.270639 -68.342514 | Washington Slagbaai National Park | Bonaire | SLI | 9.58±0.14 | 10.61±0.92 | U/Th (BON-33-BI.2, Theca) | 129.7±1.7 |
| 3681 | 12.524347 -81.729865 | San Andrés "Southwest Cove" | Colombia (San Andres y Providencia) | SLI | 1.5±0.5 | 4.5±2.06 | Other (Ge 72, 3769A) | >MIS 1 |
| 3682 | 12.556155 -81.731978 | San Andrés "May Cliff" | Colombia (San Andres y Providencia) | SLI | 6±0.5 | 12±4.03 | Other (Ge 72, 4109) | >MIS 1 |
| 950 | 13.321004 -81.387253 | Providencia Island "South Point" | Colombia (San Andres y Providencia) | SLI | 3±1.5 | 13±10.11 | U/Th | 118.8±35.64 |
| | | | | | | | Other (Ge 92) | >MIS 1 |
| 3683 | 13.324392 -81.376752 | Providencia South point | Colombia (San Andres y Providencia) | SLI | 1.8±0.5 | 21.8±20 | U/Th | 118.8±35.64 |
| | | | | | | | Other (Ge 72, 4110 a) | >MIS 1 |
| | | | | | | | Other (Ge 72, 4110 b) | >MIS 1 |


## 5.1 Brazil

Studies describing the marine deposits in Brazil date back to the late 1800s (Hartt and Agassiz, 1870). In the early 1970s, the study of Quaternary coastal deposits began with Suguio and Petri (1973) describing the Iguape-Cananéia lagoonal region at the border between the regions of São Paulo and Paraná (Figure 3). Later, the stratigraphic units of Bahia State were analyzed by Bittencourt et al. (1979) and further by Martin et al. (1982) and Bernat et al. (1983). These authors gathered new information on past sea-level changes and their meaning in the context of tectonic deformations. In the 1980s and 1990s, the exploration of Pleistocene deposits was extended to the states neighboring Bahia. In the south, Pleistocene outcrops were reported in Espírito Santo (Suguio et al., 1982), Rio de Janeiro (Martin et al., 1986; Martin et al., 1998), São Paulo (Suguio and Martin, 1995), and on the southern border of Brazil at Rio Grande do Sul (Villwock, 1984; Poupeau et al., 1988). To the north, studies focussed on the states of Sergipe (Bittencourt et al., 1983), Alagoas (Bittencourt et al., 1983; Barbosa et al., 1986), and Pernambuco (Martin et al., 1986; Dominguez et al., 1990). Most published papers presented data related to the Quaternary transgressive history of Brazil, describing the so-called "Penultimate Transgression" (called "Cananéia Transgression" in São Paulo state), attributed to MIS 5e (~120 ka). The main highstand of this transgression was reported at an elevation of ca. 8 m above sea level (a.s.l.). In general, the study of the last interglacial in Brazil is hindered by the small number of reliable chronological constraints. Therefore, the most recent studies were directed to use radiometric dating techniques (such as OSL or TL) to establish radiometric ages for Pleistocene deposits (Barreto et al., 2002; Buchmann and Tomazelli, 2003; Suguio et al., 2003; Tomazelli and Dillenburg, 2007; Rossetti et al., 2011; Suguio et al., 2011; Bezerra et al., 2015).

The collective effort from these researchers over the years has made possible the knowledge of the Brazilian coastal plain geomorphological history and the description of the Pleistocene sea-level changes, preserved mostly in the form of beach and coastal deposits. According to the literature, the last interglacial sequences are present almost continuously on a North-South gradient from Rio Grande do Sul to Rio Grande do Norte, leaving only eroded remains in the most northern states (Figure 3). Listed below are the published descriptions of the LIG deposits in this country, divided by administrative units (states).

### 5.1.1 Rio Grande do Sul

Villwock (1984) and Tomazelli et al. (2006) described a system of Pleistocene lagoons-barriers sub-parallel to the modern coast throughout the Rio Grande do Sul coastal area. Among these barriers (named I, II, and III), Barrier III is the best-preserved and has an almost continuous extension between the cities of Tramandaí (to the north) and Chuí (to the south). This barrier was associated with the LIG transgression because it occurs at the back of the Holocene lagoon-barrier system (Tomazelli et al., 2006). Tomazelli and Dillenburg (2007) re-assessed the age and elevations of Barrier III deposits at Osorio, describing five outcrops of foreshore sands with abundant *Ophiomorpha* ichnofossils (RSL IDs: 143 and 152 to 155) (Figure 3). The deposits are 4 – 5 m thick and their reported elevations refer to the maximum elevation of *Ophiomorpha* burrows,

ranging from 5.13 ± 0.7 m to 7.72 ± 0.5 m above Mean Low Water Springs (elevations measured with differential GPS). The authors recognize that there are limited chronological data for these outcrops, but highlight that a minimum age for the foreshore deposits is available for one of the sites, where coastal dunes covering the foreshore sands were dated with TL to 85 ka (Poupeau et al., 1988). Buchmann and Tomazelli (2003) used TL to date a similar foreshore deposit at Bujuru (Conceição Lightouse) to 109 ± 7.5 ka. One photo in Dillenburg et al. (2009) (their Figure 3.16) shows that this outcrop is located possibly a few meters above modern sea level. In our literature survey, we were not able to find further details on the luminescence age reported by this study or on elevation measurements of this outcrop, therefore we do not include it in our database.

### 5.1.2 Santa Catarina

In Santa Catarina State, deposits correlated with "Barrier III" deposits of Rio Grande do Sul (lagoon and barrier facies) were reported in a series of 1:100.000 geomorphological maps (Horn Filho et al., 2014 and references therein). The deposits are mapped as widespread across the coastal plain. A geological field trip guide by Horn Filho et al. (2017) describes Upper Pleistocene lagoonal/beach deposits at three sites: Villa Conceição (RSL ID: 1288), Coqueiros (RSL ID: 1286), and Tapera (RSL ID: 1287). These three outcrops are located at elevations of 9-12 m a.s.l., but their elevations are bounded by large uncertainties as it is unclear how they were measured (Figure 3).

More accurate elevation measurements are available in recent work by Martins et al. (2018). These authors investigated *Ophiomorpha* burrows within Barrier III deposits at Pinheira (RSL ID: 178). They used differential GPS to measure the top of *Ophiomorpha* at 4.5 m a.s.l (Figure 3).

Another site where beach / shallow marine (occasionally with *Ophiomorpha* burrows) deposits occur is São Francisco do Sul Island, located on the Northern coast of the Santa Catarina State (Horn Filho and Simó, 2008). These deposits were reported at 10-17 m a.s.l. (RSL ID: 179) but were assigned, in our database, as low quality due to uncertainties in their location and elevation. A detailed map (1:90.000) of the coastal deposits in São Francisco Island shows the distribution of Pleistocene lagoonal and beach deposits in this area (Horn Filho and Vieira, 2017). Accurate elevation measurements of these units will help to shed light on the correlation of these deposits with other sea-level indicators in the Santa Catarina State.

Summarizing the sandy marine terraces on the Santa Catarina states coast, Martin et al. (1988) report three additional sites attributed broadly to MIS 5: two at Itapema and one at Itajaí (RSL IDs: 1296 to 1298). These are located at 6-8 m a.s.l. (Figure 3).

Approximately 8 km south of Imbituba, Giannini et al. (2007) used OSL to date deposits associated with "*alluvial-eolian deflation*" and "*eolian accumulation*" facies. These deposits yielded ages of 129.1 ± 15 ka and 103.5 ± 11.7 ka respectively. The deflation facies is located at a lower elevation (6.5 m a.s.l) than the accumulation facies and was inserted into WALIS (RSL ID: 1300) as a terrestrial limiting point.

### 5.1.3 Paraná

In the State of Paraná, Branco et al. (2010) report a Pleistocene barrier at elevations between 5 m and 10 m a.s.l. Among 19 stratigraphic sections, they report the presence of *Ophiomorpha* burrows at Section P 01.06.05 (RSL ID: 181) at 5.5 m a.s.l. (Figure 3).

Angulo et al. (2002) describe a marine terrace deposited on an estuarine paleo-channel in Canal do Varadouro (RSL ID: 186) (Figure 3). The sediments are approximately 1 m thick and have an undulated lamination suggesting that they formed within an intertidal environment. The reported elevation is 4 m about the current high tide level (4.8 m a.s.l.). For both sites in the Paraná State, only minimum ages are available.

### 5.1.4 São Paulo

In the southern part of Sao Paulo State, the Pleistocene Cananéia Formation (a sandy coastal unit first described by Suguio and Petri, 1973), is reported in the Iguape-Cananeia lagoon region on Cananéia and Comprida Islands (Figure 3). The formation is capped by a member characterized by sands with *Ophiomorpha* burrows (Martin and Suguio, 1976). This formation was initially considered MIS 5 based on minimum radiocarbon ages (Martin and Suguio, 1976); an age later confirmed by OSL and TL ages of 94 ka (average age from Watanabe et al., 1997), and $81.55 \pm 4.5$ ka (average age from Suguio et al., 2003). We identified two (poorly constrained) sea-level index points on Cananéia and Comprida Islands (RSL IDs: 202, 203). Both data points are derived from Martin and Suguio (1976). From the description in the paper, we interpreted them as marine terraces. On Cananeia Island, the Cananéia Formation is located between 5-6 m and 9-10 m a.s.l. On Comprida Island, the altitudes vary from 2.5 m to 3 m a.s.l. in the south and from 5 m to 6 m a.s.l. in the north. As this difference is related to differential erosion to which the area was subject during the Holocene transgressive phase, we used the highest occurrence of the terrace as reported elevation.

A cross-section in Martin et al. (1988) reports another sea-level indicator associated with the Cananéia Formation close to Icapara (RSL ID: 1299) at 8.5 m above high tide (Figure 3).

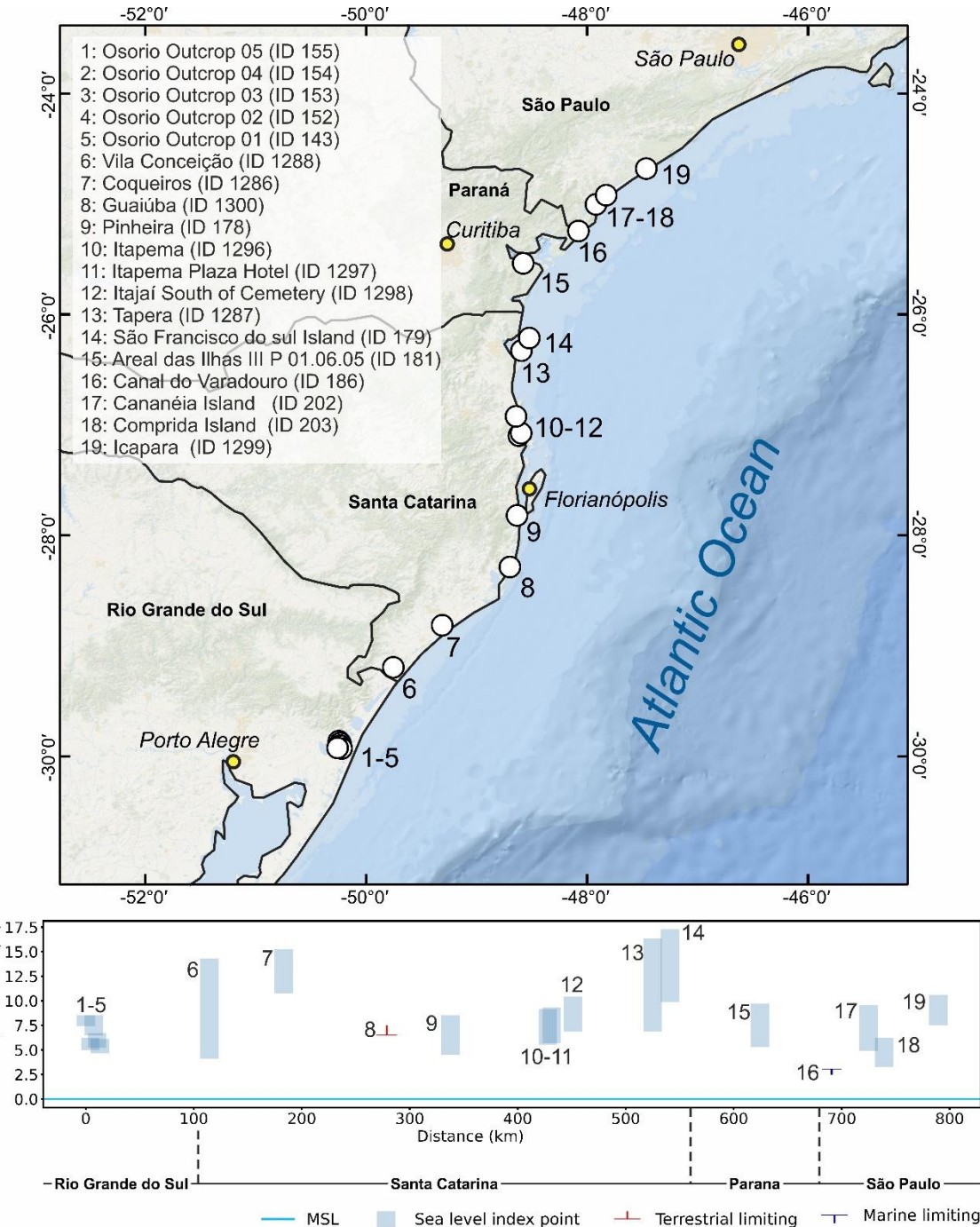

**Figure 3. Last interglacial sea-level data in the Brazilian states of Rio Grande do Sul, Santa Catarina, Paraná and São Paulo. Upper panel: map of sites. Basemap: Esri, Garmin, GEBCO, NOAA NGDC, and other contributors. Lower panel: distance/elevation plot.**

### 5.1.5 Rio de Janeiro

Martin et al. (1998) describe Pleistocene beach barriers and sandy terraces in Rio de Janeiro state. These are located at 6-8 m a.s.l. (Isla and Angulo, 2016), and have been assigned to the LIG based on infinite radiocarbon ages. As no details on sea-level
indicators are available for these terraces, we did not insert them in WALIS.

### 5.1.6 Espírito Santo

Suguio et al. (1982) studied the area near the Doce River mouth in Espírito Santo State. The authors describe the Pleistocene marine terraces which created an almost continuous strip of 4 km along the north section of the coastal plain. In the São Mateus area, these marine terraces reach a height of 9 m to 10 m a.s.l datum (RSL ID: 204) (Figure 4), while to the south (close to the
river entrance) the elevation ranges from 6 m to 10 m and the terrace loses its continuity as a result of the erosive effect from the Holocene transgression. The authors did not present specific ages for these deposits but assume a stratigraphic correlation with those of the neighboring state Bahia (see below), which indicates deposition during the LIG.

### 5.1.7 Bahia

In the State of Bahia, Martin et al. (1982) identified a fossil coral reef at "Fazenda Jarir" at an elevation corresponding to the
modern high tide mark (1.27 ± 1 m a.s.l.; RSL ID: 168) (Figure 4). They sampled 15 *Siderastrea* spp. corals; most likely, the species *Siderastrea stellata*, which is endemic to the coasts of Brazil and is reported as a primary bioconstructor in the shallow-water reefs in this area (Laborel, 1970). *S. stellata* is common in intertidal pools (de Oliveira Soares et al., 2017) with an average living range constrained between -3 m and -10 m depth (Segal and Castro, 2000). We used these values as, respectively, upper and lower limits of the indicative range to calculate paleo RSL for this site at 7.8 ± 3.6 m. These corals
yielded (alpha-counting) U-series ages between 116 ± 6.9 ka and 142 ± 9.7 ka.

Another three sites in Bahia state are reported by Bittencourt et al. (1979): Conde, Palame, and Subaúma (RSL IDs: 169, 170, 171) (Figure 4). At these sites, the authors report that "*the remnants of the penultimate transgression are indicated by a sand terrace, the top of which is situated 6 m above high tide level*". These deposits are associated with the so-called "Bahia Sector I" stratigraphy, which is attributed to MIS 5e thanks to the ages of Martin et al. (1982).

### 5.1.8 Sergipe and Alagoas

To complement the work carried out by Bittencourt et al. (1979), Bittencourt et al. (1983) analyzed the Pleistocene marine terraces deposited in the Sergipe and south of Alagoa states. According to the authors, these sandy marine terraces present the same sedimentological and geomorphological characteristics as those observed in Bahia (Section 5.1.7). Therefore, the deposits can be inferred as spatially continuous from Bahia to Sergipe and Alagoas states (Barbosa et al. 1986). While the
interglacial terraces are presented in maps within these publications, no precise location information is given, therefore these data were not inserted in WALIS.

### 5.1.9 Pernambuco

The Pleistocene marine terraces in the state of Pernambuco were described by Dominguez et al. (1990). Their elevations range from 7 m to 11 m above the present high tide level, some outcrops show traces of ancient beach ridges in the region between Lagoa Olhos-d'Agua and Boa Viagem (RSL ID: 218) (Figure 4). These terraces have similar sedimentological characteristics as those described in the states of Alagoas, Sergipe, and Bahia, suggesting a depositional continuity, however, in Pernambuco are mostly present in small patches, arranged discontinuously along the coast. There are no absolute ages for the region, but these deposits are correlated with those in the State of Bahia (Dominguez et al. 1990).

There are indications of additional MIS 5 marine-associated deposits within the Pernambuco State. Suguio et al. (2005) and later Suguio et al. (2011) reported MIS 5 TL and OSL ages for sands that could be either marine or aeolian in origin. Pending further clarifications on these deposits, we did not insert them into WALIS. On the island of Fernando de Noronha, aeolianites within Unit I of the Pleistocene Caracas Formation (Almeida, 1955) returned minimum radiocarbon ages of 50,000 years B.P. (Angulo et al., 2013). The poor constraint on age and location has precluded entry of these deposits to WALIS.

### 5.1.10 Paraíba

Suguio et al. (2005), and Suguio et al. (2011) described Late Pleistocene marine terraces in Paraíba state and dated them using TL and OSL techniques. Information to derive index points is given only for three of the nine dated outcrops. Two samples were collected above 4.1 ± 1.5 m a.s.l. at Pitimbu Beach (Figure 4) (RSL ID: 220) yield TL ages of 101 ± 9 ka (PB17A) and 71 ± 7.7 ka (PB17B) and OSL ages of 100 ± 11 ka (PB17A) and 46 ± 4 ka (PB17B). These samples were collected from a massive sandstone unit, overlying a planar cross-stratification in sandstone. As no further details are given in the original papers, we interpret these sediments as part of a marine terrace and assign this datapoint a large indicative range. A unit composed of loose sands was dated at Cordosas beach (RSL ID: 222), and yielded TL ages of 88.9 6 ka (PB07B) and 110 ± 6.2 ka (PB07C) and OSL ages of 70.3 ± 5 ka (PB07B) and 86 ± 5 ka (PB07C). The best described among the outcrops of Suguio et al. (2011) is the one at Cabo Branco cliff (RSL ID: 221) (Figure 4). Here, at 9.8 m a.s.l., a sandstone facies is characterized by planar cross-stratification and *Ophiomorpha* burrows. This deposit yielded a TL age of 138 ± 5 ka and an OSL age of 120 ± 2 ka (sample PB10B). A sandstone unit immediately above the location where these samples were taken (sample PB10A) was dated 108 ± 8 ka (TL) and 110 ± 20 ka (OSL).

### 5.1.11 Rio Grande do Norte

Barreto et al. (2002) described two distinct marine terraces in the state of Rio Grande do Norte. The sediments of these terraces were grouped into two stratigraphic units, dated to 220-206 ka and 117-110 ka with luminescence techniques. The younger 117-110 ka marine terrace deposit is preserved for about 120 km along the E-W coast and was associated with the highstand of MIS 5e. Two outcrops of this terrace, between São Bento and Touros, were described (RSL IDs: 144,163). The elevations of the shallow-water facies reported by Barreto et al. (2002) range from 1-10 m a.s.l. and 2 km north of the town of Zumbi rise

to a maximum of 20 m (Figure 4). Therefore, the authors suggested a regional tectonic uplift by 10-12 m (considering the mean reported MIS 5e highstand of Brazil: 8 ± 2 m a.s.l.; Barreto et al., 2002).

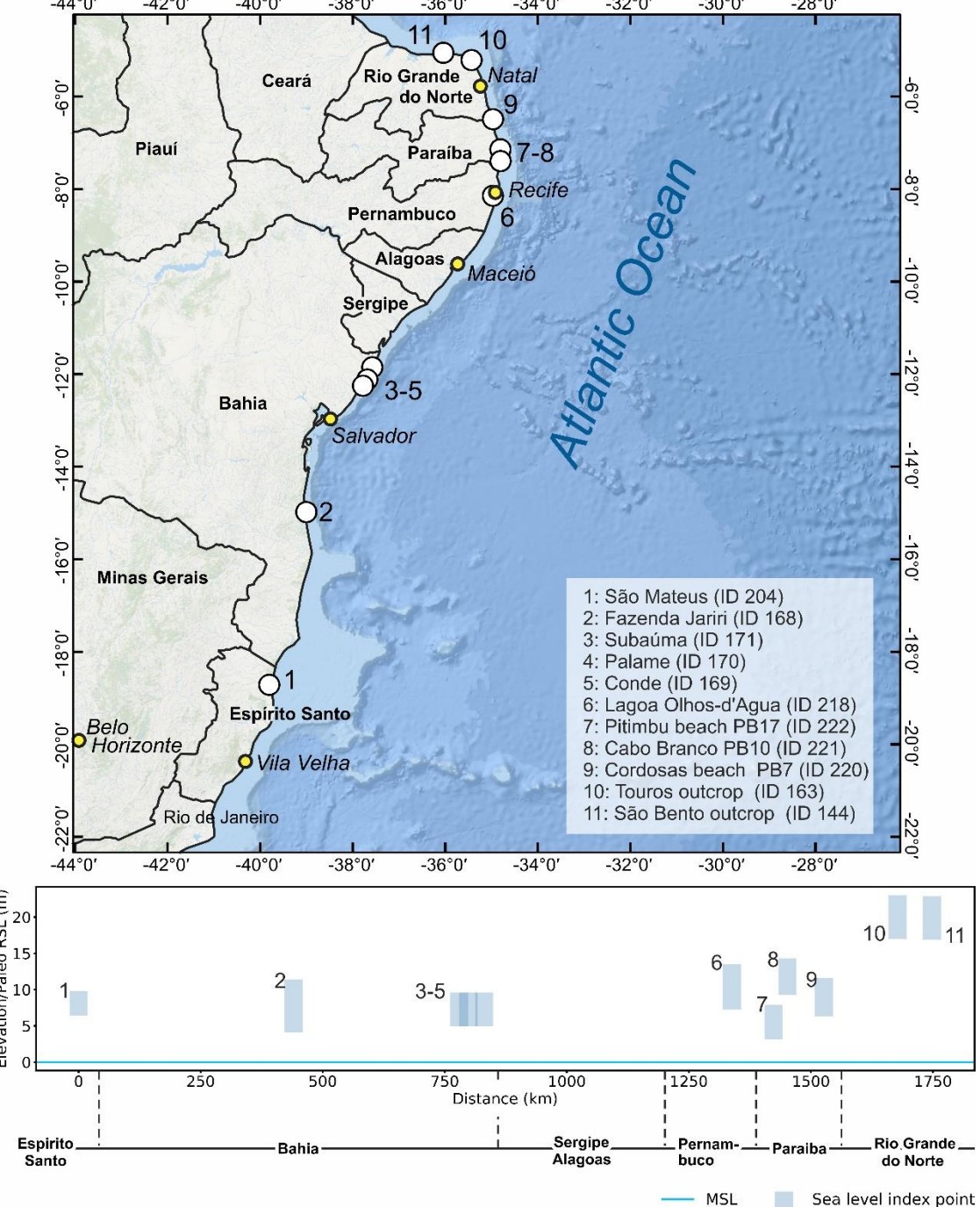

**Figure 4. Last interglacial sea-level data in the Brazilian states of Espírito Santo, Bahia, Pernambuco, Paraíba and Rio Grande do Norte. Upper panel: map of sites. Basemap: Esri, Garmin, GEBCO, NOAA NGDC, and other contributors. Lower panel: distance/elevation plot.**

### 5.1.12 Amapá

In the northernmost states of Brazil, information on Pleistocene marine deposits was generally lacking until the recent work of Bezerra et al. (2015) in Amapa State. These authors define the Pleistocene "Itaubal Formation", and subdivide it into two progradational units (Upper and Lower), separated by an unconformity. Using OSL ages, the Itaubal Lower Unit was constrained to MIS 5 (120 ka to 71 ka). Detailed facies analysis along several outcrops allowed Bezerra et al. (2015) to assert that the Itaubal Lower unit is representative of "*subtidal and tide-influenced meandering stream and floodplain deposits*".

While this study presents accurate chronological data and detailed facies analysis of the Lower Unit, it is not possible to insert any index point in the database due to the lack of any absolute elevation measurements constrained to modern sea level (only outcrop thickness is reported). However, we remark that this area looks promising for future research, especially because of the limited number of LIG deposits preserved in northern Brazil.

### 5.2 French Guiana, Suriname & Guyana

French Guiana, Suriname, and Guyana share a very similar geological setting. Along the coasts of these three countries, different authors report the presence of littoral deposits that were emplaced by previous sea-level highstands (Brinkman and Pons, 1968; Iriondo, 2013). These deposits were studied by Choubert (1956) and Boyé and Cruys (1961), who described a unit made of sands and clays parallel to the shoreline that was initially attributed to the Riss-Würm (MIS 5) transgressive event. In French Guiana and Suriname, similar sedimentological facies are named, respectively, "Coswine Series" and "Coropina

Series". Brinkman and Pons (1968) propose that the Coswine and Coropina Series are divided into two members, the Para member (attributed to the middle Pleistocene) and the Lelydorp member (attributed to the Eemian interglacial, MIS 5e). These authors suggest that the Lelydorp member outcrops between the towns of Cayenne and Organabo in French Guiana and Suriname in the district of Coronie.

The attribution of the Lelydrop member to MIS 5e was due to a radiocarbon date above the detection limit considered as a

minimum age by Brinkman and Pons (1968) (48,000 years B.P., sample ID: GRN 4718). Wong (1992) continued with the study of this region publishing "*the Quaternary stratigraphy of Suriname*", in which he addressed the problem related to the chronological assignment of these eroded and weathered records. Wong et al. (2009) used paleomagnetic data to estimate ages along the Suriname coastal plain. The results suggest that the Lelydorp member is of early Pleistocene age, hence much older than hitherto assumed. Due to the lack of precise chronologic constraints, and with the work of Wong et al. (2009) essentially

pre-dating the units previously assumed to be of last interglacial age, no data has been inserted in WALIS for French Guiana, Guyana, and Suriname.

## 5.3 Venezuela

The Pleistocene marine deposits of Venezuela are well-known and have been extensively described by Bermudez and Farias (1975). As early as the late 1700s, Humboldt (1799) remarked upon formations now recognized as Pleistocene-age within the
state of Sucre (Bermudez and Farias, 1975). A couple of centuries later, Bermudez (1969) presented a detailed account of the Quaternary and recent stratigraphy of Venezuela. In this study, the author mentions Pleistocene marine units on Cabo Blanco (Miranda State), on the south coast of Tortuga Island, and the islands Cubagua, Coche, and Margarita. In Cabo Blanco, Bermudez (1969) describes Pleistocene "*raised beaches*", located at 62 m a.s.l. Danielo (1976) worked on the Northern coasts of Venezuela, and reported the presence of several Pleistocene beach deposits in the Araya and Paraguana
Peninsulas, as well as in Puerto Cumareboy and Margarita regions. Among the different sea level proxies on the northern coasts of Venezuela identified by the author, it appears that two would correlate with MIS 5, notably the so-called "Tyrrhénien I" (25-30 m) and the "Oujien" (6-8 m). Unfortunately, there is no dating associated with these deposits, therefore they could not be included in our database.

The Paraguana Peninsula is one of the most studied sites in Venezuela for which concerns Quaternary outcrops. Here, Rey
(1996), described a 1.7 m-thick conglomeratic sequence containing fragments of mollusks and foraminifera. The author interprets the depositional environment as that of a high-energy beach and reports that this unit cannot be older than the Pleistocene due to the presence of the marine foraminifera *Globorotalia truncatulinoides*. Audemard (1996a,b) reported several Pleistocene coastal outcrops along the Paraguana Peninsula. On the southern coast of the peninsula, at Punta Cardon, Audemard (1996a,b) report the presence of a fossil coral reef, with a height of 1.5 m and species of the genera *Porites*
preserved in living position within a "*reddish sand matrix*". The authors attribute this reef tentatively to MIS 5. To the west of the Paraguana Penninsula, Audemard (1996a) reports a terrace with heights from 4 m to 5 m a.s.l. This terrace presents sediments with different grain sizes and fragments of shells and corals. One radiocarbon analysis was performed on a coral fragment; its age was above the detection limit and the author assigns an MIS 5 age. To the north of Paraguana, the same author describes two eroded "*isolated beach deposits*" at Punta Macolla, despite not presenting an elevation for these, he
correlates them with the MIS 5.

After the work of Audemard (1996b), no further references could be found in this review. Up to now, none of the studies listed above present radiometric dates, and the estimated ages were based on either geomorphological (Danielo, 1976; Audemard, 1996a, b; Rey, 1996) or biological (Bermudez, 1969; Bermudez and Farias, 1975) characteristics of the deposits described. Only the work of Audemard (1996a) refers to a single coral radiocarbon sample with age beyond the dating limit. None of the
studies listed was included in WALIS.

### 5.4 Aruba, Curaçao, and Bonaire (ABC) Islands

Aruba, Curaçao, and Bonaire islands lie in front of Venezuela, forming the so-called "ABC Islands" group. These three islands are included in this review due to the proximity to the Venezuelan coast and the quantity of last interglacial sea-level indicators that have been reported along their shores. The Pleistocene sea-level record at these islands is mostly preserved in the form of staircases of coral reef terraces (Alexander, 1961; Herweijer and Focke, 1978). These are usually wider along the windward side of the island than their leeward sides. The terraces display shore-parallel changes in elevation, which would suggest they have been affected by tectonic processes. This seems likely as they are located between the South Caribbean and South American Plates (Hippolyte and Mann, 2011). The reef terraces are often interrupted by "Bocas", i.e. incisions in the continuity of the Pleistocene reefs that expose the stratigraphy of the terrace, which can be also observed along sea cliffs. These outcrops facilitated the study of paleo-ecological properties of Pleistocene reefs (Pandolfi and Jackson, 2001; Meyer et al., 2003), their dating (Schellmann et al., 2004; Obert et al., 2016; Felis et al., 2015; Schubert and Szabo, 1978; Hamelin et al., 1991), and helped unravel their significance in the context of paleo relative sea-level changes (Lorscheid et al., 2017; Muhs et al., 2012; Kim and Lee, 1999). The evidence of MIS 5e reef terrace development on each island is briefly described hereafter.

### 5.4.1 Aruba

In Aruba, the so-called "*third terrace*" (third terrace level counting from modern sea level) was attributed to a "*Sangamonian*" (i.e., MIS 5e) age by Alexander (1961). The terrace is reported as well developed and is parallel to the shore along the North-Western coasts of the island. The terrace elevation is "25 feet" (7.6 m) a.s.l. While Pleistocene marine terraces appear prominent in the landscape of Aruba, we could not find any description nor absolute ages to confirm the MIS 5e age designation; therefore, we did not insert any datapoint for Aruba in the database.

### 5.4.2 Curaçao

Several last interglacial MIS 5e index points were reported from the island of Curaçao (Figure 5). The first U-series ages on corals from Curaçao were reported by Schubert and Szabo (1978) and later revisited by Muhs et al. (2012). We included in WALIS the sites reported by Schellmann et al. (2004), analyzed with ESR, and those reported by Muhs et al. (2012), measured with a differential GPS and analyzed with U-series. In reporting U-series and ESR data for Curaçao, we included only those ages within MIS 5e. We did not insert in the database the U-series ages obtained by Hamelin et al. (1991), as they were rejected by the original authors due to: i) large difference in ages (multiple ka) between different subsamples of the same coral or, ii) outside the acceptable range for initial uranium isotopic composition.

In general, within the "Bocas" dissecting the lower terrace of Curaçao, there are two distinct units where well-preserved corals (often in growth position) appear: the Cortalein (lower) and Hato (upper) units (Figure 6). Both ESR and U-series ages confirm that the Cortalein unit was forming during MIS 7 (ca. 200 ka), while the Hato unit is MIS 5e in age (Muhs et al., 2012; Schellmann et al., 2004). We report in the database only the dated samples collected from the Hato unit. In the following

paragraphs, we report sample IDs as indicated in Table 2 of Schellmann et al. (2004) and Table 1 of Muhs et al. (2012) (for U-series).

The elevations reported by Muhs et al. (2012) are referenced to the CARIB 97 geoid (Smith and Small, 1999). We assigned an elevation uncertainty of 0.95 m, calculated from the root mean square of the sum of squares of the maximum error reported by Muhs et al. (2012) (0.8 m) and the CARIB97 datum uncertainty (0.51 m). Elevations for the sites mentioned in Schellmann et al. (2004) were taken directly from their Table 2, were referred to a general "mean sea level" datum, and were assigned an arbitrary uncertainty of 20% of the measured elevation. A similar approach was used in reporting sites investigated by Schubert and Szabo (1978).

On the leeward (southeastern) side of the island, a reef sequence at Punta Halvedag (RSL ID: 3553) was reported up to 10 m above present sea level. For this sequence, we included in WALIS four ages reported by Muhs et al. (2012) on two corals (Cur-Dat-16 and Cur-Dat-17). Muhs et al. (2012) was concerned about the U values for seawater and considered the ages to be overestimated by ~2.5-3.5 ka. Therefore, in WALIS we constrain the Punta Halvedag site as "younger than" these two corals, with the caveat that their age is most likely MIS 5e. North of Punta Halvedag, the same reef sequence is visible at Knipbai (RSL ID: 3554). As little information on the stratigraphic context is given for this site, we inserted it in WALIS as a marine limiting datapoint. From the Hato Unit at this location, one *Acropora palmata* was dated (Cur-Dat-5; Muhs et al., 2012). Muhs et al. (2012) report that this coral "*shows evidence of U gain, which would tend to bias the sample to a younger apparent age*". Therefore, in WALIS we constrain the Knipbai site as "older than" this coral.

On the windward (Northwestern) side of the island, most of the locations reported in WALIS have been surveyed inside "Bocas", where corals have been dated both with ESR and U-series. At Un Boca, three coral samples (99-6, 99-7, 99-9) yielded ESR ages between $101 \pm 7$ ka and $140 \pm 9$ ka (Schellmann et al., 2004). The top elevation of in situ corals at Un Boca is 10 m with significant uncertainties that stem from the unreported elevation measurement method. At Un Boca (RSL ID: 531), Muhs et al. (2012) re-dated two corals, that had already been dated to MIS 5e by Schubert and Szabo (1978). The new analyses yielded ages of $118.1 \pm 0.8$ ka (Cur-33-d, *Acropora palmata*) and $132.3 \pm 0.8$ ka (Cur-33-d, *Diploria* sp.). The authors note that the sample Cur-32-d is probably biased old by 2.5-3.5 ka. One *Acropora palmata* coral (original ID: 00-7) was analyzed by Schellmann et al. (2004) at a site located a few hundred meters south of Un Boca, called Dos Bocas (RSL ID: 534). Three subsamples of this coral yielded ESR ages between $113 \pm 10$ ka and $120 \pm 11$ ka. The reef terrace at Dos Bocas was assigned the same elevation as Un Boca, based on the data reported by Schellmann et al. (2004). Slightly more than 1 km south of Dos Bocas, another site (Boca Mansalina, RSL ID: 3559) yielded one *Acropora palmata* coral (sample 00-13) analyzed using ESR to provide an age of $118 \pm 9$ ka (Schellmann et al., 2004) and one *Siderastrea siderea* coral (sample Cur-Dat-4), of which two subsamples gave U-series ages of $125.7 \pm 0.7$ ka and $126.0 \pm 0.8$ ka. The highest *in situ* coral at this location is reported at 7-8 m a.s.l. and it appears to have been sampled close to the top of the terrace (Figure 8 panel 3 of Schellmann et al. (2004) and Figure 8a of Muhs et al. (2012). *Acropora palmata* corals were also dated with ESR (00-5) and U-series (Cur-Dat-1) at Boca

Cortalein (RSL ID: 533), yielding ages between 118-125 ka (with error bars between 8-11 ka) and U-series average age of 127.3 ka (on two subsamples of the same coral). Also, at Boca Cortalein, we approximate the height of the reef terrace with the highest in situ coral, which is reported at 10 m a.s.l. (Schellmann et al., 2004). Approximately 15 km to 18 km south of Boca Cortalein, three additional sites were reported by Schellmann et al. (2004) and Muhs et al. (2012). At Boca Ascension (RSL ID: 3563), a *Montastraea* sp. coral yielded an ESR age of 124 ± 8 ka (99-3; Schellmann et al., 2004), but not enough stratigraphic information was given to establish a sea-level index point, therefore we insert this site in WALIS as marine limiting. At Boca San Pedro (RSL ID: 535), four *Acropora palmata* corals (00-9) were collected at 6-7 m a.s.l. and were dated with ESR to 103-124 ka (Schellmann et al., 2004). In this case, a cross-section (Fig.5-2 of Schellmann et al., 2004) shows that the highest of these corals was sampled close to the top of a reef terrace, therefore we consider this point a valid sea-level indicator.

Towards the southern part of the island, south of Hato International Airport, two nearby sites were dated with ESR by Schellmann et al. (2004): Boca Labadera (RSL ID: 536) and Boca Grandi (RSL ID: 537). These sites have ages ranging between 120 ± 13 ka and 112 ± 9 ka. At these two sites, all dated samples (n=5, WALIS ESR IDs 124 to 128) are close to the top of the terrace, were measured at 5-6 m above sea level, and have been treated in WALIS as valid sea-level indicators. At the southern tip of Curaçao, one site was reported by Schellmann et al. (2004) as "Sheraton Hotel" and is here reported as Oostpunt (RSL ID: 532). Here, two *Acropora palmata* corals (001-1) were sampled at 2.5-4 m. As no information is given on the stratigraphy of the reef terrace at this site, we insert this point in WALIS as a marine limiting point.

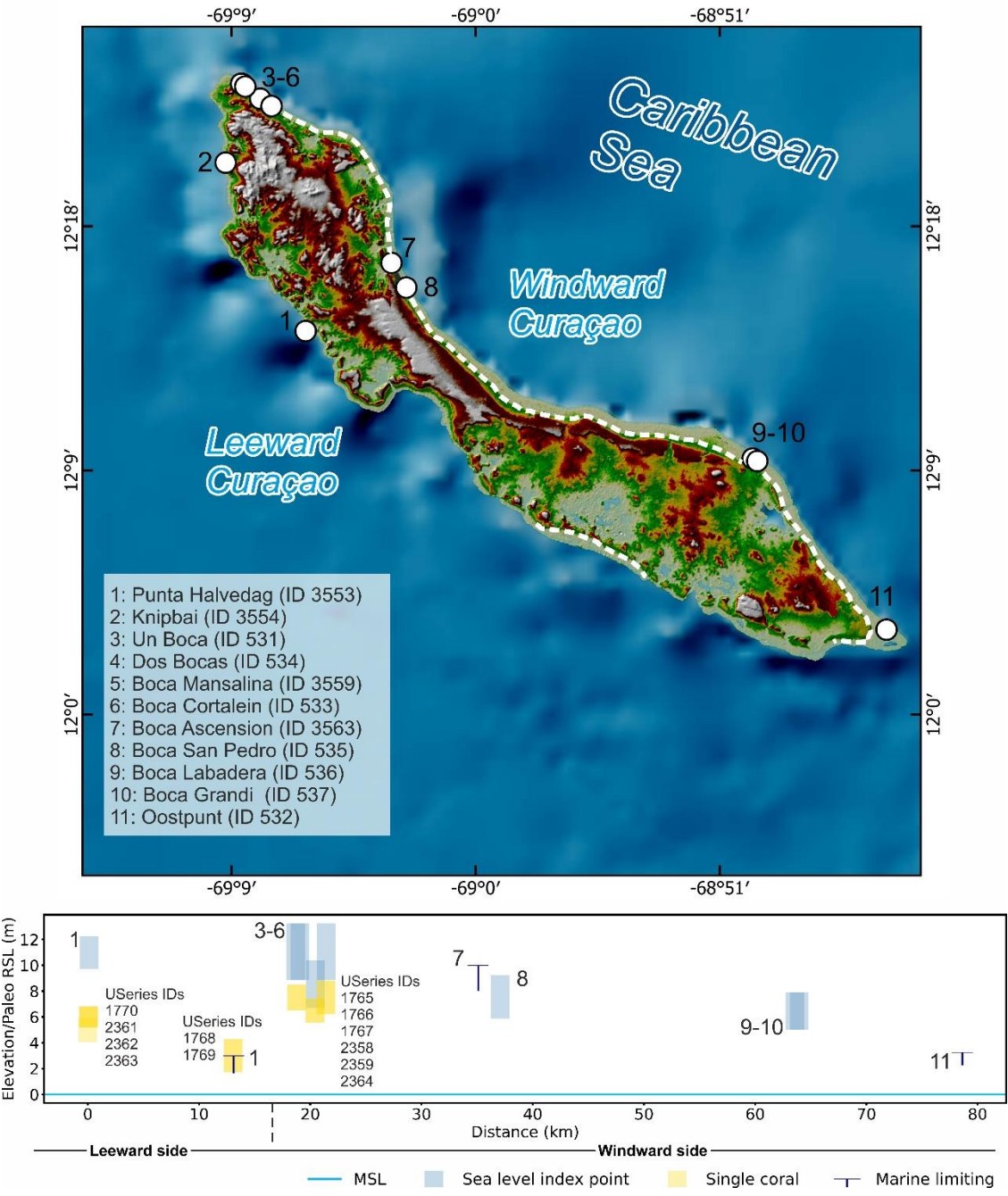

**Figure 5. Last interglacial sea-level data in Curaçao. Upper panel: Map of reported sites. The dashed line shows the location of the last interglacial terrace. The "single coral" datapoints represent coral elevations as reported in Chutcharavan and Dutton (2020). Background map compiled with data from GEBCO (doi:10.5285/836f016a-33be-6ddc-e053-6c86abc0788e), SRTM 30m by NASA EOSDIS Land Processes Distributed Active Archive Center (LP DAAC, https://lpdaac.usgs.gov/). Lower panel: distance/elevation plot.**

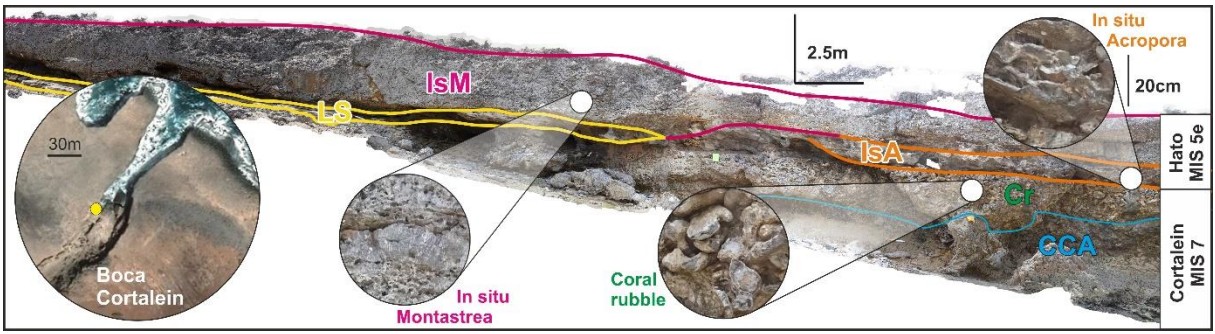

**Figure 6. Photomosaic with the interpretation of the paleo reef units of MIS 7 and MIS 5e age at Boca Cortalein, Curaçao (originally by A.-K. Petersen, edited by A. Rovere).** *IsM*: In situ *Montastraea* sp.; *LS*: Layered sediments (beachrock); *IsA*: In situ *Acropora* sp.; Cr: Coral rubble; CCA: Solitary corals and calcareous algae.

### 5.4.3 Bonaire

The general coastal setting of Bonaire, similarly to that of Curaçao and Aruba, is characterized by broad (hundreds of meters wide) paleo reef terraces on the windward (Northern and Eastern) side of the island and narrow (tens of meters or less) paleo reef terraces on the leeward (South Western) side (Figure 7).

On the leeward side, Lorscheid et al. (2017) measured six tidal notches carved into limestones older than MIS 5e (Figure 8a). The elevations of the notches were measured with a combination of differential GPS and laser rangefinder and are reported between 6.61 m and 7.26 m (RSL IDs: 1369 to 1374) a.s.l. Samples of a fossil coral from the terrace immediately below the notches yielded initial $^{234}U/^{238}U$ activity ratios higher than expected from the modern seawater value and were therefore considered unreliable by Lorscheid et al. (2017). Nevertheless, the notches are considered coeval with the terrace immediately below (Figure 8b), which is correlated with the better-dated terrace level on the windward side of the island, described below.

Felis et al. (2015) and Obert et al. (2016) report several U-series ages from different skeletal parts (theca walls or bulk material) of nine corals located on top of the lower reef terrace characterizing the windward (northern and eastern) side of Bonaire (Figure 8c, d, e). As there are multiple sub-samples for some individual corals, Brocas et al. (2016) calculate the weighted mean and weighted standard error of five of these nine corals giving a range from 120.5 ± 1.1 ka to 125.85 ± 2.46 ka. Of the 42 ages reported by Obert et al. (2016) and one age reported by Felis et al. (2015), we inserted in WALIS only those accepted within the original publication (based on progressively less strict criteria), restricting the number of available ages to 25 (8 corals). The elevation of these samples was initially measured using an altimeter "calibrated" to sea level at the time of measurement. The sampling sites were then re-visited by T. Lorscheid, A. Rovere, and T. Felis in 2016, and each sampled coral on top of the reef terrace was re-measured with differential GPS and referred to the EGM 2008 geoid. We report these new measurements in this paper for the first time. To provide additional constraint to each sample, an elevation was also recorded at the nearest accessible instance of the coral reef inner margin, which we consider here to approximate paleo sea

level with an indicative range included between Mean Lower Low Water and the Breaking Depth, that we derived from IMCalc (Lorscheid and Rovere, 2019).

South of Boca Washikemba, Obert et al. (2016) and Felis et al. (2015) dated two corals: BON-5-A (120.5 ± 1.1 ka average age
according to Brocas et al., 2016), and BON-5-D (117.7 ± 0.8 ka). Three kilometers north, the inner margin of the reef terrace was measured at 5.19 ± 0.28 m, where *in situ* massive corals can be recognized (RSL ID: 694). While this is the only site dated on the eastern coast of Bonaire, the wide coral reef terrace is almost continuous until a second site is reached on the northeastern coastline (identified within WALIS as Boca Olivia, RSL ID: 693), where Obert et al. (2016) dated two other corals BON-26-A, and BON-24-AII.2 giving average ages of 124.9 ±19 ka and 125.5 ± 2.4 ka respectively (Brocas et al., 2016).  The inner
margin of the coral reef terrace was measured 500 m inland of these two corals, at an elevation of 8.84 ± 0.27 m. Three kilometers north of this point, south of Boca Onima (RSL ID: 692), three corals (BON-17-AI, BON-12-A, and BON-13-AI.1) yielded an average age of 124.3 ± 1.5 ka (Obert et al., 2016; Brocas et al., 2016). The inner margin correlated to these corals was measured very close to the sample locations, at an elevation of 5.72 ± 0.3 m. Four kilometers north of Boca Onima close to the Washington Slagbaai National Park border (RSL ID: 3472), Obert et al. (2016) dated one coral (BON-33-BI.2) to 129.7
± 1.7 ka. The inner margin of the terrace closest to this coral was measured at 9.58 ± 0.14 m. The measured elevations show a trend to become higher towards the North. That the coral ages reveal a broad trend from younger at lower elevations in the South to older at higher elevations in the North, with intermediate ages at intermediate elevations in between, has been noted by Obert et al. (2016) and attributed to the slight tilting of the island with relative uplift in the north and submergence in the south (Hippolyte and Mann, 2011). Consequently, this trend is unlikely to reflect past sea-level variations (Obert et al., 2016).

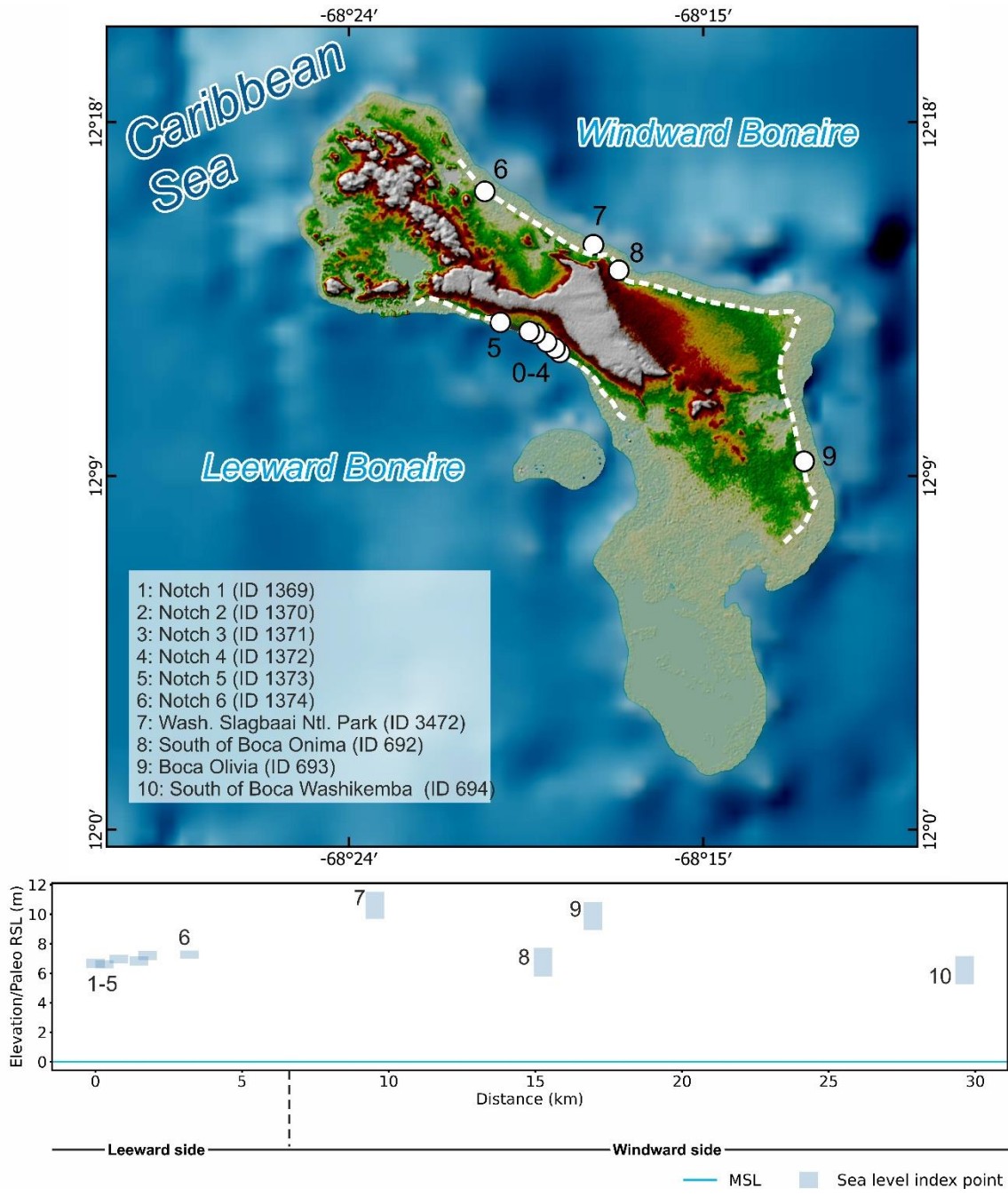


Figure 7. Last interglacial sea-level data in Bonaire. Upper panel: map of sites. The dashed line shows the location of the last interglacial terrace. Basemap compiled with data from GEBCO (doi:10.5285/836f016a-33be-6ddc-e053-6c86abc0788e), SRTM 30 m by NASA EOSDIS Land Processes Distributed Active Archive Center (LP DAAC, https://lpdaac.usgs.gov/). Lower panel: distance/elevation plot.

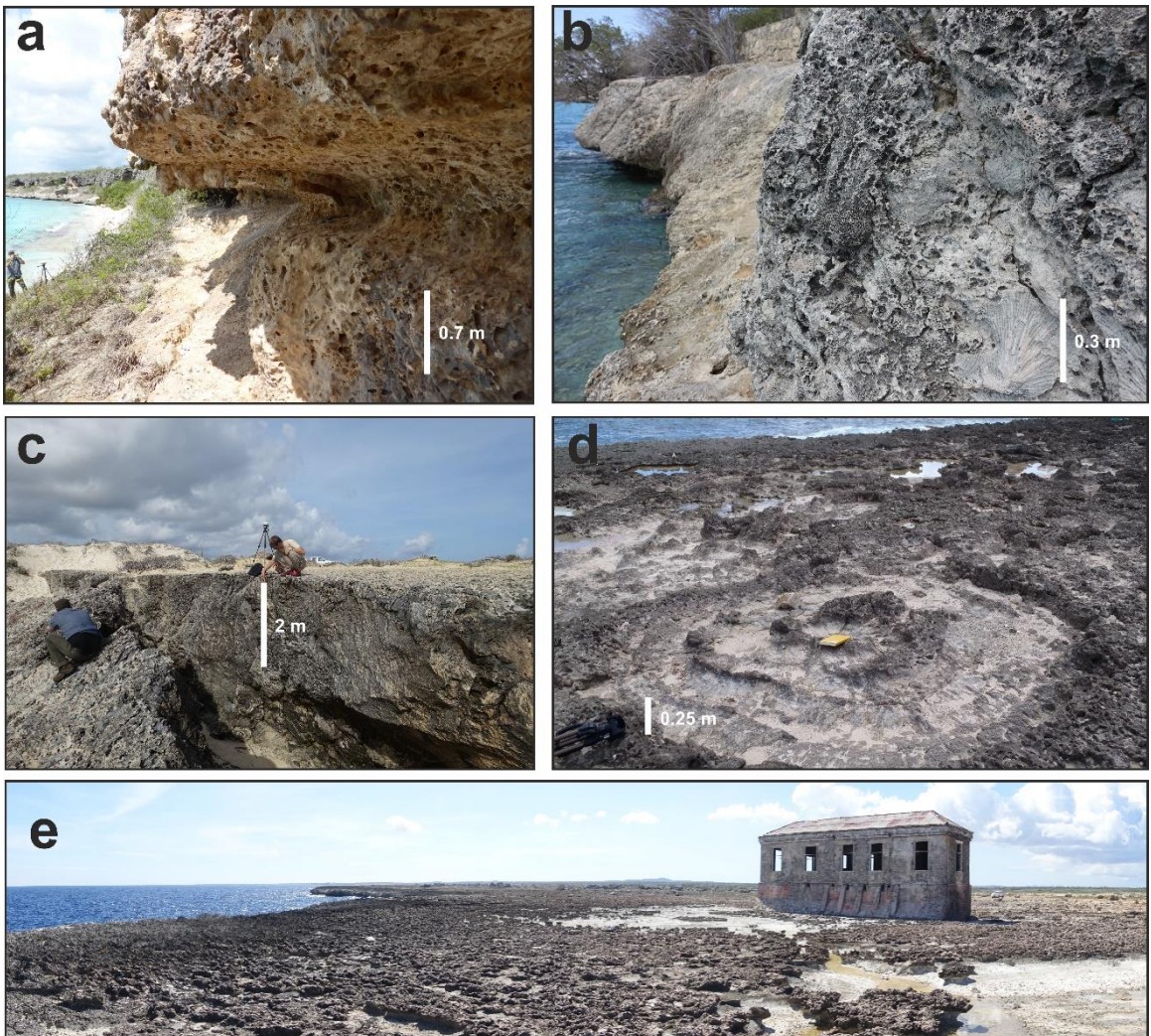


**Figure 8. a)** Tidal notch on the leeward side of Bonaire; **b)** last interglacial reef 1-2 meters below the location of the photo shown in a); **c)** exposed reef section on the windward side of Bonaire; **d)** large massive coral eroded on top of the MIS 5e terrace on the windward side of Bonaire; **e)** panorama view of the MIS 5e terrace on the windward side of Bonaire. Photos by T. Lorscheid (panel a), A. Rovere (panels b,c,e), T. Felis (panel d).

## 5.5 Colombia to Honduras


The coastal deposits of Colombia associated with the LIG have been mainly studied and described on offshore islands (Porta and Solé de Porta, 1960; Bürgl, 1961; Geister, 1972; Geister, 1986; Geister, 1992). Porta et al. (2008) mentioned the existence of marine terraces and coral platforms of different elevations along the mainland Colombian coast, mainly between Cartagena and Barranquilla; however, no further details are given on their ages or elevations or specific locations.

Porta and Solé de Porta (1960) describe the Quaternary coastal deposits on Tierrabomba Island. While the main focus of their paper is to describe the marine faunas of the islands, the authors mention Pleistocene marine terraces with heights of more than 20 m; however, none of these deposits are directly correlated with the LIG. One year later, Bürgl (1961) mentions three geological and geomorphological units on San Andrés Island: i) a marine platform of recent age, ii) a terrestrial platform of Pleistocene age, and iii) inland limestones of Miocene age. Geister (1972) hypothesizes that the deposits on the "terrestrial"

platform were formed during a marine transgression, and correlates them with coral terraces observed in Providencia Island (Figure 9) according to its geomorphology. Geister (1986) suggests the Providencia terraces are of Sangamon Interglacial age (MIS 5) based on a minimum radiocarbon age. The same author (Geister, 1992) later described this and the rest of the coral deposits of Providencia. He highlights that this fossil reef terrace is the only emerged relict of the Pleistocene complex which now underlies the Holocene deposits.

Literature describing MIS 5 relative sea-level indicators in Panama, Costa Rica, Nicaragua, and Honduras is scarce. Only one study in Costa Rica mentions the presence of a paleo-coral reef for which an Eemian age (MIS 5e) is postulated, in Puerto Viejo (Bergoeing, 2006). We surmise that it is possible that MIS 5 outcrops have not yet been described or do not exist in Panama, Nicaragua, and Honduras. For Panama and Costa Rica, several studies focus on the Pacific zone (Bee, 1999; Davidson, 2010; Bauch et al., 2011), which is out of the area of interest for this paper. No studies have been found in Nicaragua

and Honduras, but we do not discard the possible existence of descriptions of MIS 5 outcrops in journals that we could not access online.

### 5.5.1 San Andrés Island

From the work of Geister (1972), it is possible to derive two sea-level indicators (generally defined as reef terraces) on the island of San Andrés. One is located in the south of the island at a point called "Southwest Cove" (RSL ID: 3681) and the

second to the north, in "May Cliff" (RSL ID: 3682) (Figure 9). The elevation of the "Southwest Cove" site was 1.5 m a.s.l., and the coral sampled belonged to the species *Acropora palmata*, which has an average living range between -1 m and – 5 m depth (Lighty et al., 1982). We use these values as, respectively, the upper and lower limit of the indicative range to calculate the paleo RSL for this site at 4.5 ± 2.06 m (RSL ID: 3681). This sample was radiocarbon dated, giving an age of 26,020 ± 675 years B.P., which according to the author should be considered as a minimum age. The "May Cliff" sample comes from

a *Dendrogyra cylindurs* coral (from -2 m to -10 m living range in San Andrés, as reported by Cavada-Blanco et al., 2016) found at 6 m a.s.l. with a minimum radiocarbon age of 33,000 ± 770 years B.P. Considering the average living range of *D. cylindurs* (Cavada-Blanco et al., 2016) on this island, the paleo RSL calculated is 12 ± 4.03 m (RSL ID: 3682).

### 5.5.2 Providencia

To the south of the island of Providencia, Geister (1972) reports the elevation and two radiocarbon dates (33,310 ± 2300 years

B.P., and 29,270 ± 930 years B.P.) of a specimen of the coral *Siderastrea radians*. The reported elevation of the sample is 1.8

m a.s.l. and both radiocarbon dates were considered as minimum ages. Twenty years later, the same author describes a coral reef terrace at about +3 m at a site called "*South Point*" (RSL ID: 950) (Figure 9). One U-series age confirms the association of these deposits with MIS 5e, albeit with a large error (118 ± 11.0 ka; U-SERIES ID: 2026) (Geister, 1992).

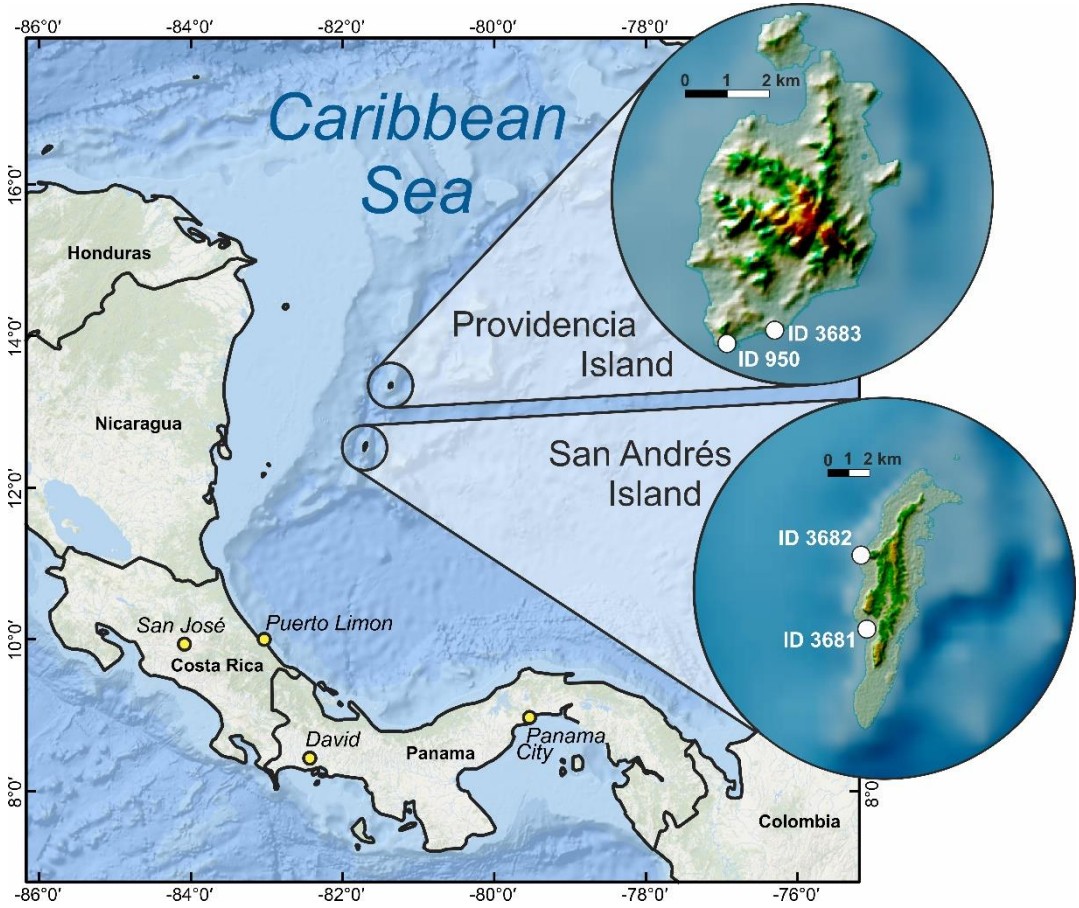

**Figure 9. Location of Providencia and San Andrés islands within the Caribbean Sea. The insets show last interglacial sea-level data on the two islands. Basemap: Esri, Garmin, GEBCO, NOAA NGDC, SRTM 30m by NASA EOSDIS Land Processes Distributed Active Archive Center (LP DAAC, https://lpdaac.usgs.gov/).**

## 6 Further remarks and conclusions

### 6.1 Data quality

Each RSL datapoint in our compilation has been assigned two quality scores, one for age and one for RSL information. The quality ranking goes from 0 (rejected) to 5 (excellent) and follows the guidelines given in Rovere et al., 2020. Thanks to the standardized WALIS interface, similar scores are available also for RSL datapoints located to the South and to the North of our region, which were compiled into WALIS by Gowan et al. (2021) and Simms (2021). In Figure 10, we compare the quality

scores assigned in our review to those assigned by these studies. This comparison shows that the quality of both age and RSL information for our study area is, in general, higher than those in the nearby sites. However, the high scores on both properties are driven by the sites in the ABC islands, and no site in Brazil goes above the 'Average' age scores. For which concerns the RSL information, in Brazil only the sites described by Tomazelli and Dillenburg (2007) and Martins et al. (2018) have been scored as 'Good' to 'Excellent'. This means that, potentially, these sites have final uncertainties on paleo RSL below two meters but their attribution to MIS 5e should be supported by more reliable dating.

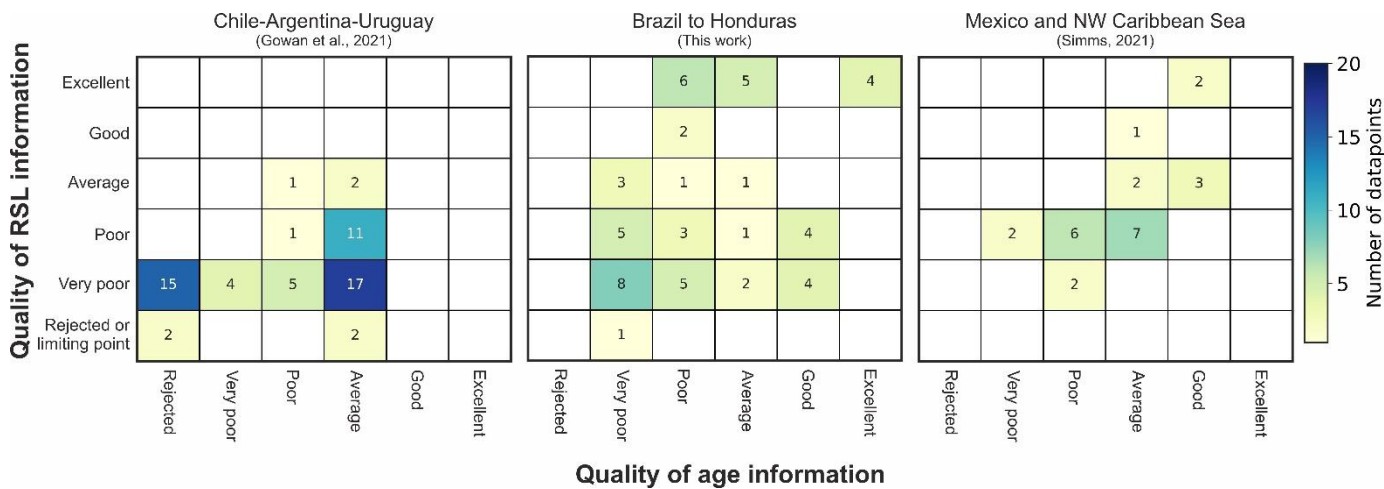

Figure 10 Quality of RSL and age information as estimated by (from left to right) Gowan et al. (2021) for Chile-Argentina-Uruguay, Brazil-Honduras (this study), and Mexico and NW Caribbean Sea (Simms, 2021). All studies used the same rationale to give a score to age and RSL information (see Rovere et al., 2020).

## 6.2 Departures from the eustatic signal

While in the WALIS interface there is the option to compile data and metadata on tectonic rates that affect each reported RSL data point, we chose to leave these fields blank. The reason behind this choice is that late Quaternary tectonic rates along the coasts covered by our review are often back-calculated from the elevation of the last interglacial shoreline and assumptions MIS 5e eustatic sea level, and therefore inserting such rates in the database would be affected by circularity. However, we remark that the area is characterized by different tectonic settings. The southern portion of Brazil (Figure 3) is located roughly in the center of the South American Plate, hence sits on a passive margin. Towards the North (Figure 4), several authors noted an increase in seismicity and highlighted the presence of faults offsetting Neogene deposits (Bezerra et al., 2006).

As remarked in the regional descriptions above, in this area last interglacial deposits appear higher than 20 meters and are hence considered uplifted (Barreto et al., 2002). The ABC islands are located on the Caribbean Plate, hence the elevation of MIS 5e sites on these islands is affected by tectonic displacement. Based on several levels of Quaternary reef terraces, Muhs et al. (2012) calculated that sites on the island of Curaçao are characterized by uplift rates in the order of 0.026-0.054 m/ka. Similarly, the island of Bonaire has also been affected by tectonic uplift, which appears to have caused a tilting of the island.

Lorscheid et al. (2017) report that this tilting is 192 mm/km to the southeast, with the direction of the tilting comparing well with the one reported by Hippolyte and Mann (2011).

Besides tectonics, other regional processes might influence the elevation of RSL indicators in the area of interest. Earth Dynamic Topography has been shown to cause significant departures from eustasy in last interglacial sea-level records also along passive margins. Along the coasts of Brazil, Earth Dynamic Topography might contribute to several meters of uplift (Austermann et al., 2017, their Figure 3A, C). However, these predictions are still bounded by significant uncertainties (Austermann et al., 2017, their Figure 3B) and cannot be employed to 'correct' the RSL records shown here.

Another significant factor that may have caused the displacement of last interglacial RSL datapoints is sediment isostasy (Pico, 2020), which is defined as the isostatic response to sediments deposited by large rivers on the shelf. In particular, this process may affect areas close to the Amazon river (Pico, 2020, their Figure 3) causing net land subsidence in the order of tens of meters. This could explain why we did not find studies describing last interglacial RSL data points in this area, hinting that they might be located well below modern sea level.

A further process that has surely affected the current elevations of last interglacial RSL sea-level proxies along the coasts of Brazil to Honduras is glacial- and hydro-isostatic adjustment (GIA), which is caused by the isostatic response of the Earth to mass fluctuations of continental ice sheets. GIA consists of both solid Earth and mean sea surface (i.e. geoid) vertical variations that show up as relative sea-level (RSL) variations. The deviation of the local GIA-modulated RSL changes from the global mean, i.e. "eustatic", sea-level change, depends primarily on the distance from the ice sheets. During glacial periods such as MIS 6, the crust subsides underneath a growing ice sheet in response to the ice load. The upper mantle material is therefore pushed outwards to accommodate the bending lithosphere. At the same time, the increased mutual gravitational pull between the ice mass and the ocean water, causes the latter to rise in the proximity of the ice margins. As a result, ice-proximal locations experience RSL rise, which is significantly larger than the eustatic fall. Further away from the ice margins, the crust undergoes uplift in response to the upper mantle flow that is directed radially outwards and upwards. This is the so-called uplifting peripheral forebulge area, where the local glacial RSL drop is significantly larger (and faster in time) with respect to the global eustatic drop. Moving away from the forebulge, the local RSL drop tends to approximate the global eustatic signal, with deviations that depend on the water-loading redistributions in response to solid Earth, gravitational as well as rotational perturbations.

During deglacial and interglacial periods such as the MIS 5, the GIA process operates in the same way, but the overall trend switches. Accordingly, RSL drop is expected over the previously ice-covered region, while the collapse of the uplifted peripheral forebulge results in RSL rise larger than the eustatic. Interestingly, during the termination of the deglacial phases, the far-field areas that are either along the continental margins or in the equatorial band comprised between the tropics, do experience a noticeable RSL fluctuation that consists of an early highstand (above the eustatic value), then followed by a drop. Two processes are at work here: (i) the so-called continental levering, which consists of an upward tilt of the continental

margins in response to the subsidence of the ocean basins which are loaded with meltwater, and (ii) the ocean syphoning, where water migrates towards the collapsing peripheral forebulges in response to ocean mass conservation.

## 6.3 Last interglacial sea-level fluctuations

The current research status of the last interglacial in the Western Atlantic and Southwestern Caribbean does not allow for the discernment of sea-level oscillations within MIS 5e, since most studies in the region lack the refined chronology necessary to identify such detailed sea-level patterns. However, we remark that this area might help to solve several questions related to the presence and magnitude of sea-level oscillations during MIS 5e, if high-quality data will become available.

To illustrate this point, we model the GIA-modulated RSL changes by solving the gravitationally self-consistent sea-level equation (SLE; Spada and Stocchi, 2007). We use a solid Earth model that is spherically symmetric, self-gravitating, rotating and is divided into shells that are characterized by a linear Maxwell viscoelastic rheology. We divide the mantle into three layers and imposed a vertical stratification of viscosity following the VM2 mantle viscosity profile of Peltier (2004). We force our model with the ANICE-SELEN ice-sheets chronology, which consists of four ice sheets (North America, Eurasia, Greenland, and Antarctica) and covers the last 410 kyrs of climate fluctuations (i.e., four glacial-to-interglacial cycles; de Boer et al., 2014). For the MIS 5e interglacial, we impose a global mean sea level scenario where Greenland and Antarctic ice sheets release, respectively, 2.5 and 1.0 m ESL equivalent at 127 ka. The Greenland ice sheet remains stationary until 117 ka, while the Antarctic ice sheet releases another 4.5 m after 120 ka. This scenario is in line with the 'two-stepped' last interglacial sea level proposed by Hearty et al. (2007) and O'Leary et al. (2013), which is still debated as little evidence supports the notion of a rapid collapse of the Antarctic Ice Sheet (AIS, Barlow et al., 2018; Polyak et al., 2018).

The GIA model shows a strong latitudinal dependence of the RSL change along the transect composed by our compilation and those of Gowan et al. (2021) to the South and that of Simms (2021) to the North (Figure 11b). Such a regionally-varying RSL pattern depends on the interplay between the collapsing forebulge in the north and the continental levering and ocean syphoning in the south.

In particular, at the northernmost site of XCaret, the predicted RSL elevation at 127 ka is still ~6 m lower than the eustatic value (i.e., +3.5 m a.s.l.). This stems from the local contribution of the collapsing forebulge that was uplifted in response to the North American ice sheet glaciation at the MIS 6. The viscous response of the upper mantle results in a delay of the local RSL change, which reaches, through a monotonic rise, the actual eustatic value only at 120 ka, i.e. right before the final jump that is caused by the AIS melting. A similar trend, but smaller in amplitude, is predicted at Boca Cortalein, thus implying that the forebulge-related processes cease to exist between Venezuela and northern Brazil, where a true eustatic 'sea-level jump' is expected. The predicted RSL curve at Sao Bentos-Touros is close to the eustatic, although the combination of levering and syphoning is visible in the form of a ~1.0 m highstand at 127 ka, which is then followed by an RSL drop towards the eustatic value at 120 ka. The GIA-driven highstand at 127 ka increases towards the south and reaches its maximum at Camarones, i.e.

at a significant distance from the North American ice sheet and Antarctica. Interestingly, the predicted highstand at 127 ka is of the same magnitude as the final peak between 119 and 117 ka. Under such scenario, there should be evidence, along the entire coasts of Argentina and central Brazil, of a high-to-low sea-level swing, caused by the interplay between GIA and eustatic changes. At Puerto Williams, instead, the proximity to the AIS results in a much lower highstand at 127 ka, most likely as a function of the reduced gravitational pull after the MIS 6 and 5e melting.

The effects of solid Earth and gravitational perturbations are also visible in the predicted elevations of the final jump that is caused by the AIS fast melting. The southernmost sites, being closer to the AIS, are affected by the geoid drop that is caused by the reduced gravitational pull. As a result, the predicted highstand is slightly lower than the eustatic. On the other hand, the northernmost sites experience a higher-than-eustatic peak, again as expected by the self-gravitation process.

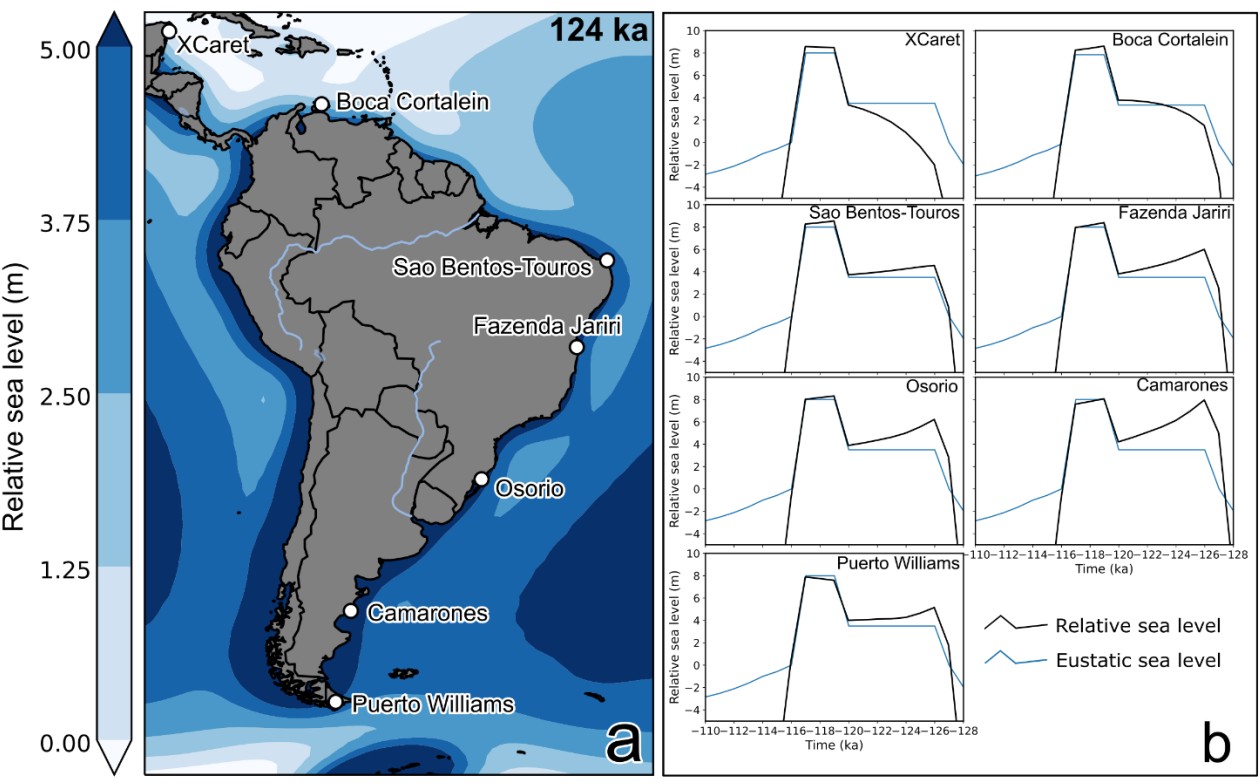

**Figure 11 a) Relative sea level as predicted by the ANICE-SELEN model at 124 ka along a transect including selected sites in our review and those of Gowan et al. (2021) and Simms (2021); b) Time vs RSL at each location shown in a).**

**6.4 Other interglacials and Holocene sea-level indicators**

Among the studies reviewed, there are some reported ages/inferences related to other Quaternary sea-level highstands. In Brazil, Barreto et al. (2002) and Suguio et al. (2011) describe shorelines with elevations ranging from -2 m to 10 m a.s.l. that

were associated with MIS 7 (substage 7c). In Curaçao, Schellmann et al. (2004) and Muhs et al. (2012) associate the Lower Terrace with the MIS 7 estimating a paleo sea level between -3.3 m to + 2.3 m, and the Middle Terrace with MIS 11, estimating paleo sea level ranging between approximately +8.3 m to +10 m.

For the Holocene, the work of Khan et al. (2017) presents standardized sea-level index points in Honduras, Panama, Venezuela, Curaçao, Guyana, and Suriname. A review of Holocene studies in Brazil was conducted by Angulo et al. (2006) and a database
is available, albeit not yet standardized to state-of-the-art templates for Holocene sea-level studies (i.e., Khan et al., 2019).

## 6.5 Future research directions

There are several lines of inquiry that merit attention for which concern LIG sea-level studies in the area covered by this database.

1. Except for the islands of Bonaire and Curaçao, the age control on sea-level index points associated with MIS 5 is
generally poor. Most locations have limited chronological control, relying upon minimum radiocarbon ages or chronostratigraphic correlations between sites. In Brazil, more accurate chronological techniques have been employed to date Pleistocene sediments (i.e. OSL and TL), but there is a general lack of reporting standards of geochemical values and metadata, which makes it difficult to assess the reliability of each sample.

2. With few exceptions (Tomazelli and Dillenburg, 2007; Martins et al., 2018), the elevation of Brazilian sea-level
proxies should be better measured with state-of-the-art techniques, such as differential GNSS systems, to reduce the currently large uncertainties. Therefore, a research priority for the vast area between Rio Grande do Sul and Rio Grande do Norte (Figures 3 and 4) is to perform new fieldwork aimed at re-measuring and re-dating the sites reported in our database, providing enough data and metadata on both elevation and age.

3. In Venezuela, the Paraguana Peninsula appears as a potential target for investigations on LIG shorelines. Here, there
is a general need for better site descriptions, location, and measurement. Coupled with U-series and/or OSL ages at selected sites, this will enable the possibility to add to our database several sea-level index points. For which concerns French Guiana, Suriname, and Guyana, there is evidence that LIG transgressive sequences are preserved, but the central need is to identify key sites and provide reliable geological descriptions, ages, and elevation measurements. Similarly, in the long stretch of coast from Colombia to Honduras, a research priority for future studies is to identify
whether last interglacial sites are present. The fossil coral reef in Puerto Viejo (Costa Rica) reported in Bergoeing (2006) may represent the starting point for investigations on LIG sea-level changes in the region, which might also encompass a better description and dating of the reefs in the San Andrés and Providencia Islands.

4. In Bonaire and Curaçao, the sea-level index points are generally well-described, precisely measured, and dated. This stands in strong contrast with the neighboring area of Aruba, where reef terraces are only generally reported in the
literature, but for which age control and stratigraphic descriptions are not available. Completing the LIG history of the "ABC islands" by including details on the fossil reefs of Aruba appears as a priority in this area.

5. Together with the data compiled in Simms (2021) and Gowan et al. (2021), the data in this paper provide a unique transect across the forebulge of the former Antarctic Ice Sheet. The location of this transect may be used to test different ice melting histories, including testing the possibility of a rapid Antarctic Ice Sheet collapse during the last interglacial. This will be possible only by largely improving, with new field data, the data quality both on RSL index points and on their associated age.

## 7 Data availability

The Western Atlantic and Southwestern Caribbean database is available at: https://zenodo.org/record/5168571 (Version 1.02; Rubio-Sandoval et al., 2021). The description of the database fields can be found at: https://zenodo.org/record/3961544 (Rovere et al., 2020).

## Author contributions

KRS compiled the database and wrote the MS with help from AR and DDR. AR, TL, TF, CC, AKP and PS contributed original field data from Bonaire and Curaçao. All authors gave input on the manuscript, revised the final text and agree with its contents.

## Competing interests

The authors declare no competing interests.

## Acknowledgments

This work is part of the PhD thesis of Karla Rubio-Sandoval, funded by the ERC Starting Grant "WARMCOASTS" (ERC-StG-802414). We would like to thank Peter Augustinus, Edward Antony, Theo Wong, Franck Audemard, Alcina Barreto, and Afonso Nogueira for clarifying aspects of the RSL indicators across different areas of the region reviewed. Figures 4, 3 and 9 were created using ArcGIS® software by Esri. ArcGIS® and ArcMap™ are the intellectual property of Esri and are used herein under license. Copyright© Esri. All rights reserved. For more information about Esri® software, please visit www.esri.com. The data used in this study were compiled in WALIS, a sea-level database interface developed by the ERC Starting Grant "WARMCOASTS" (ERC-StG-802414), in collaboration with PALSEA (PAGES / INQUA) working group. The database structure was designed by A. Rovere, D. Ryan, T. Lorscheid, A. Dutton, P. Chutcharavan, D. Brill, N. Jankowski, D. Mueller, M. Bartz, E. Gowan, and K. Cohen. The data points used in this study were contributed to WALIS by Karla Rubio-Sandoval and Alessio Rovere, with Peter Chutcharavan assisting with some U-series data entry.

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
