# Peer review of "A review of Last Interglacial sea-level proxies in the Western Atlantic and Southwestern Caribbean, from Brazil to Honduras"

_Earth System Science Data, 2021_

## Author Comment (AC1)

**Responses to the Comments of the Reviewers**

**Manuscript No. essd-2021-150**

**A review of Last Interglacial sea-level proxies in the Western Atlantic and Southwestern Caribbean, from Brazil to Honduras**

Dear Editor, we would like to thank you for your work and three reviewers for their constructive comments. Please see attached a new version of the MS with track changes, and a point-by-point response to the reviewers' comments below. We are confident that the modifications suggested by the reviewers helped us to improve the MS, and we look forward to hear from you in due course.

In the following, our text is in blue ink, while the original reviewer's comments are in black ink.

**Responses to the comments of Reviewer #1 Thomas M. Cronin (RC1 comments)**

**Citation: https://doi.org/10.5194/essd-2021-150-RC1**

Thomas M. Cronin:

General. This is a useful summary, but I am bothered by the one main conclusion in the abstract: that nothing but future research ideas came of the compilation. How about some of the more high-quality SL reconstructions? None with stratigraphy and chronology are informative? I would think the data in Table 3, with a few assumptions, could be very revealing for MIS 5 SL. And the Caribbean sites certainly have a seal level-tectonics signal, maybe GIA too.

We thank the reviewer for the honest assessment of our data compilation. We try not to do any assumptions on the different sites, as we want to keep the paper as descriptive as possible. This is also contained in the guidelines of the journal and the special issue, which require an honest assessment of data but with limited interpretation. However, as we feel it is important to give the reader at least a feeling for both tectonics and GIA, we modified Figure 1 to include general tectonic plates and data from the global faults database. We also included a new section discussing departures from eustasy (Section 6.2) and a new figure (Figure 11) showing the output of a selected GIA model. This should give, in our opinion, enough perspective on this topic.

Given the large literature from for ex. Brazil, isn't there a relative SL record from the best studied and dated areas? Could such reconstructions be re-illustrated, re-interpreted in this paper?

It is very difficult to accurately identify studied and dated areas from Brazil. Several sites are identified in the literature, but often their description is scattered through different studies and were completed before it was common to provide accurate stratigraphic sections and reference to modern sea level. The best stratigraphic evidence is probably the one presented by Tomazelli and Dillenburg (2007), but they show the stratigraphic sections with photographs, so it is hard to reconcile these into a stratigraphy. We prefer to refrain from making interpretations based upon inaccurate data, and we do hope that our review will encourage other scientists to pick up the work where we left it.

On what basis is the white line along the coast in Figure 1 presumed to have relict shorelines? It seems like it just follows the coast? Related to this, the compilation really [and admittedly] uses a whole lot of different shoreline indicators, each having varying quality and methodology. See Tables 1 and 2.

The white line along the coast was supposed to indicate places where MIS 5 shorelines could be located but not enough information is available to create SL index points. Thanks to this comment, we second-guessed our drawing choice. We now realize that it confused the reader, so we took it out. It is indeed true that the compilation uses very different indicators, and that is reflected in our text. We leave to the database user the choice of filtering out data according to dating technique / indicator or quality.

Moreover, without adequate chronology and mapping, who knows if some are deposits or geomorphic features are not early Pleistocene? Pliocene, even Miocene?

That is correct, and this is the reason why the new Figure 1 does not include the white line. We feel that the discussion of these locations and our conclusion establish that more work is needed in these areas before SL index points are extracted for the LIG.

As you read the text on regional studies, there seems inconsistency about selection criteria to be included in WALIS, some areas with undated features are included, some with dates only have some included.

We tried to clarify better, wherever possible, the reasons for including or excluding sites in our compilation.

Finally, it seems unusual not to have discussions of GIA and tectonics, which is likely found in parallel papers from other coasts in the MIS 5 sea level volume

ESSD requests data papers, presenting data without too much room for discussions. However, we see the point of the reviewer: reminding the reader about tectonics, GIA and other processes is important. We therefore expanded the discussion of these processes in section 6.2 and 6.3.

Specific

Line 13. I'm not sure what this means: "assigned to one or more geochronological constraints"

This sentence was changed to: "each constrained by one or more geochronological methods"

Line 18. Or this, "to identify sea-level index"

To be clearer, we add the definition of index point in brackets:" discrete past position of relative sea level in space and time".

Or line 29-30: "to insert standardized sea- level points for several areas"

This sentence was changed to: "to allow a proper standardization of sea-level data for the remain coastal areas".

Line 53 says: "we extracted 50 sea-level index points" but abstract says 55, are these synonymous?

In the abstract we stated that we produced 55 data points, this value includes both index points and limiting points (50 index points, 4 marine limiting points and 1 terrestrial limiting point). A sea-level index point defines the discrete position of past relative sea level in space and time, whereas limiting data provide an upper (terrestrial limiting data point) or lower (marine limiting data point) bound on the past position of relative sea level at a given point in space and time.

Figure 1. The "white dashed line" is really a series of small dots? Confusing with the circles.

White line was deleted.

Line 92 what is a "used a total station to measure"

A total station is an instrument commonly used in topography to measure elevations. https://en.wikipedia.org/wiki/Total_station

Line 103 reword: "For which concerns the geographic positioning of sites"

This sentence was reworded

Line 123 reword: "thanks to stratigraphic similarities"

This sentence was reworded

Line 224. Fix this, you mean 94 ka right? "94,504 ka"

Yes 94 ka is right, we made the modification according to your suggestion.

Figure 6 seems out of place in this data compilation paper.

As we had this sketch from our own surveys, we thought that it might have been interesting for the reader to see at least one outcrop characterization. We refer to the editor on this: if he thinks that this figure is not necessary, we can drop it without problems.

Line 520. Aren't there many studies of Neogene [possibly with Quaternary terraces] along the Caribbean coast of Costa Rica?

As mentioned in the text, the only study we could find for the Caribbean coast of Costa Rica is the one by Bergoeing, 2006, but there is not enough metadata in that study to create an index point for WALIS.

Line 558. Rewrite this: "For which concerns the Holocene"

This sentence was reworded

**Responses to the comments of Reviewer #2 (RC2 comments)**

**Citation: https://doi.org/10.5194/essd-2021-150-RC2**

Referee 2:

This Is an interesting dataset that screened and reviewed indicators along the coasts of the Western Atlantic and Southwestern Caribbean, on a transect from Brazil to Honduras that includes the islands of Aruba, Bonaire, and Curaçao.

The work summarises 55 standardized datapoints, each assigned to one or more geochronological constraints from a variety of relative sea-level indicators including beach deposits, coral reef terraces, marine terraces, burrows, and tidal notches. Like many in this volume and in recent years the paper focuses on concerns related to age control and the accuracy of elevation measurements.

The work then concludes rather flatly with a bland finale that much more is to be done. While I agree here I think much has been done and I am fairly sure there are sea level records from Brazil that could be compared to. I was also left feeling how does this study site compare to others in the volume. How does this dataset stand up against others with more or better data?

We thank the reviewer for this comment. To answer it, we inserted a specific section in the "Further remarks and conclusions" section, where we discuss about the data quality scores we assigned during the review. We compared this score with the score of all the other data already published in WALIS, in order to put it in perspective. We hope that adding such part to the discussion will answer this concern.

I have one primary criticism of the work and that relates to the discussion or lack thereof regarding tectonics. The authors skim over the tectonics of the region and it is almost certain that parts of the study area ie. **Netherlands Antilles** that will have been subject to tectonic contamination. A good starting point is maybe Wang et al., Remote Sens. 2019, 11(6), 680; https://doi.org/10.3390/rs11060680. Perhaps a section could be added on how to decontaminate or otherwise address sites that are clearly affected by tectonics either past or present.

We agree with the reviewer that some discussion on tectonics was missing in the previous version of the MS. To answer this query, we now updated Figure 1 and added a section (6.2) to explain the difference between the Brazilian shelf and the areas further to the North, on the Caribbean plate. In the same section, we discuss other processes causing departures from eustasy, such as Dynamic Topography, sediment isostasy and GIA.

In summary - the work is publishable and provides a good summary of the dataset and in particular some well though guidelines for future work.

We thank the reviewer for the time they took to review our MS. The constructive comments helped us improving the manuscript.

**Responses to the comments of Reviewer #3 Rafael C. Carvalho (RC3 comments)**

**Citation: https://doi.org/10.5194/essd-2021-150-RC3**

Rafael C. Carvalho:

The paper by Rubio-Sandoval et al. addresses the last interglacial sea-level proxies from Brazil to Honduras. The authors review and report a total of 55 index points extracted from 36 papers. The sea level indicators which comprise 50 of the index points are identified and discussed in terms of elevation, datum, and dating techniques. Additional sections summarizing the existing knowledge re sea level and discussing controversies and potential research directions into the future are also included.

My impression when I first read this manuscript was that the authors did a good job in terms of citing the scientific literature (at least, in my case, for the more familiar Brazilian coast), a basic prerequisite for a good review. I was happy to see that a comprehensive appraisal was conducted not only for the existing literature in English but also in Portuguese. However, several issues made me wonder whether the article itself was appropriate to support the publication of this dataset, and therefore I recommend a major review for this contribution.

We thank the reviewer for the time he took in reading our MS, and for the constructive comments that helped us improve our work.

What's the rationale for this latitudinal extent covering such broad regions with different tectonic settings (e.g. between Brazil and sites in the Caribbean)? Looking at the other articles in WALIS, I see a case for covering large areas (e.g. Freisleben et al. ---most of the Pacific coast of SA), but I was wondering whether the Caribbean datasets should be independent. If not, the authors must at least acknowledge this issue in introduction.

The rationale for including such a large geographic zone was to fill a gap in currently published WALIS compilations. To the North, Simms (2021) is extending down to Mexico, while to the South, Gowan et al. (2021) is covering up to Uruguay. We explained this rationale in the introduction. It is also important to mention that due to the limited data in the Caribbean (mostly from Curaçao and Bonaire) we would not have enough data for a standalone paper.

I was really surprised to see that despite its extension, not a single topographic profile/schematic cross-section, stratigraphy, sequence of depositional events, satellite image or even a photograph of the Brazilian coast was presented. This creates a contrast when compared to what's being presented for the Caribbean (Fig 6 and 8).

This is true, and the reason is rather simple: we do not have such data for Brazil. In the ABC islands, we have original data (as stated in the MS), therefore we could grond-truth the landforms that are visible on DEMs. Not having similar datasets for Brazil, we preferred to refrain from interpretations that would be only remote, without ground truthing.

There's clear potential for this dataset to be used based on its uniqueness, usefulness and completeness. However, a statement claiming that this database contribution represents a starting point (abstract) should be avoided considering this is using secondary data. Apart from the description of the dataset, the discussion is rather limited, and this reflects in the abstract and conclusion.

Thank you, we avoided this statement in the abstract.

E.g. If discernment of SL oscillations is not possible for Brazil, how about to discuss this with the aid of the Pleistocene SL curve, highlighting the reliable data of Tomazelli and Dillenburg, 2007; Martins et al., 2018? Or to use generalised cross-sections from several sites around Brazil similar to what was presented for the Australian coast (Murray-Wallace and Belperio 1991) for another discussion topic.

For which concerns sea-level oscillations, we took this and other suggestions from the reviewers and added a new section to the discussions. Unfortunately, it is very difficult, if not impossible, to draw an updated and complete section for Brazil, at the current status of knowledge. We put this as an endeavor for future studies in the discussion section.

In terms of cartographic content, the paper lacks quality. Map figures alternate between different colour palettes representing DEMs by different colours. I suggest standardising the colour scheme throughout paper and incorporate legends. I also feel that much more could be done to Figs. 3-5, 7 and 9, which are currently limited to represent the location of samples under a range of different scales, without really adding much information to what is already presented in Fig 1. I suppose all those figures (3-5, 7 and 9) could be incorporated as inserts into a larger Fig 1. This way, the reader would have a general idea of the point distribution and also have a better understanding at a larger scale of the south, northeast of Brazil (from north Bahia to RG do Norte only), Curacao, Bonaire, Providencia/San Andres points on a single figure. Regardless of this more complex Fig 1, the other figures need to become more informative and make better use of the data compiled by the authors. A bit of cartographic skills would make figs 3-5, 7 and 9 to represent the information discussed in text. E.g. Fig 3 could be made of three side-to-side maps representing elevation, datum and dating techniques (colour symbology). If this is done also for the other figs, the reader would then benefit from understanding much more than just the spatial distribution of the index points.

The choice of using DEMs for the reef islands stems from the fact that, on these datasets, it is really clear where the coral reef terrace sits. Using similar maps for the Brazilian coast (much smaller scale compared to the ABC islands) would add, in our opinion, very little information. With respect to the suggestion of "replicating" maps within a single figure, we remark that it would create a rather redundant bulk of information, also considering Table 3 and the database annexed, that can be easily downloaded and mapped with any categorization chosen by the user. We feel that the important part of all figures is the lower panel, where the elevation pattern of RSL information is presented. It is rather difficult to incorporate all the information contained in figures 3-5,7 and 9 in a single figure 1.

Please increase font size of Fig 2 and the distance elevation plots in figs 3-5, and 7. By the way is there a reason for not having a similar plot in Fig 9? Another observed inconsistency regards the labelling of points in those maps. I suggest to stick to Wallis IDs similar to what was done in fig 9. Therefore, get rid of the 0-18 in fig 3, 0-10 in fig 4, 0-10 in fig 5, 0-9 in fig 7, and label points according to IDs. Regarding scales, Fig 9 inserts have tiny graphical scales and font sizes. The other figs especially 5 and 7 lack scales!

We increased fonts as suggested. We did not add the plot in Figure 9 as there are only few datapoints, and the distance / elevation graph would be trivial. For which concerns the labelling, we tried to do as suggested, but as WALIS IDs are rather long (up to 3 or 4 characters), they would clutter the maps too much and make everything rather hard to read and understand. We

inserted scales to field photos whenever possible, and we thank the reviewer for pointing this out.

The paper by Suguio et al 2005 also reports 12 Pleistocene TL/OSL dated samples and locations from the coast of Pernambuco and Rio Grande do Norte. 6-7 of those samples are from the MIS 5 and should at least be discussed why they are not incorporated into this review (similar to what was done with Fernando de Noronha).

We thank the reviewer for pointing out this paper, which had escaped our attention. We cross-checked the luminescence ages in Suguio et al., 2005 with those in the database, and we could verify that several are indeed already included as they were used in Barreto et al., 2002 and Suguio et al., 2011. These papers also give the stratigraphic context and elevation for these ages, which is not available for the ages we left out from Suguio et al., 2005. We, however, added this reference to both the database and the text, to make sure that any reader can track back the original sources.

[revised manuscript text omitted]